# An integrative phylogenetic approach for inferring relationships of fossil gobioids (Teleostei: Gobiiformes)

Christoph Gierl[1], Martin Dohrmann[1], Philippe Keith[2], William Humphreys[3,4], Hamid R. Esmaeili[5], Jasna Vukić[6], Radek Šanda[7], Bettina Reichenbacher[1,8]*

1 Department of Earth and Environmental Sciences, Palaeontology & Geobiology, Ludwig-Maximilians-Universität München, Munich, Germany, 2 UMR 8067 Biologie des Organismes et Ecosystèmes Aquatiques (BOREA), Muséum National d'Histoire Naturelle, CNRS, IRD, SU, Paris, France, 3 School of Biological Sciences, University of Western Australia, Crawley, WA, Australia, 4 Western Australian Museum, Welshpool, WA, Australia, 5 Ichthyology and Molecular Systematics Research Laboratory, Zoology Section, Department of Biology, College of Sciences, Shiraz University, Shiraz, Iran, 6 Department of Ecology, Charles University, Prague, Czech Republic, 7 Department of Zoology, National Museum, Prague, Czech Republic, 8 GeoBio-Center LMU, Ludwig-Maximilians-Universität München, Munich, Germany

* b.reichenbacher@lrz.uni-muenchen.de

**Data Availability Statement:** All relevant data are within the manuscript and its Supporting Information files. The data underlying the results

## Abstract

The suborder Gobioidei is among the most diverse groups of vertebrates, comprising about 2310 species. In the fossil record gobioids date back to the early Eocene (c. 50 m.y. ago), and a considerable increase in numbers of described species is evident since the middle Miocene (c. 16 m.y. ago). About 40 skeleton-based gobioid species and > 100 otolith-based species have been described until to date. However, assignment of a fossil gobioid species to specific families has often remained tentative, even if well preserved complete specimens are available. The reasons are that synapomorphies that can be recognized in a fossil skeleton are rare (or absent) and that no phylogenetic framework applicable to gobioid fossils exists. Here we aim to overcome this problem by developing a phylogenetic total evidence framework that is suitable to place a fossil skeleton-based gobioid at family level. Using both literature and newly collected data we assembled a morphological character matrix (48 characters) for 29 extant species, representing all extant gobioid families, and ten fossil gobioid species, and we compiled a multi-gene concatenated alignment (supermatrix; 6271 bp) of published molecular sequence data for the extant species. Bayesian and Maximum Parsimony analyses revealed that our selection of extant species was sufficient to achieve a molecular 'backbone' that fully conforms to previous molecular work. Our data revealed that inclusion of all fossil species simultaneously produced very poorly resolved trees, even for some extant taxa. In contrast, addition of a single fossil species to the total evidence data set of the extant species provided new insight in its possible placement at family level, especially in a Bayesian framework. Five out of the ten fossil species were recovered in the same family as had been suggested in previous works based on comparative morphology. The remaining five fossil species had hitherto been left as family *incertae sedis*. Now, based on our phylogenetic framework, new and mostly well supported hypotheses to which clades they could belong can be presented. We conclude that the total evidence framework

presented in the study are available from https://figshare.com/s/ed10b9a5ac382f856a20.

**Funding:** BR received funding from the Deutsche Forschungsgemeinschaft (grant number RE-1113/20). RS received funding from the Ministry of Culture of the Czech Republic (grant number DKRVO 2019-2023/6.III.d National Museum, 00023272). https://www.dfg.de/ https://www.mkcr.cz/ The funders had no role in study design, data collection and analysis, decision to publish, or preparation of the manuscript.

**Competing interests:** The authors have declared that no competing interests exist.

presented here will be beneficial for all future work dealing with the phylogenetic placement of a fossil skeleton-based gobioid and thus will help to improve our understanding of the evolutionary history of these fascinating fishes. Moreover, our data highlight that increased sampling of fossil taxa in a total-evidence context is not universally beneficial, as might be expected, but strongly depends on the study group and peculiarities of the morphological data.

## Introduction

The suborder Gobioidei (Percomorpha: Gobiiformes) is among the most diverse groups of fishes, encompassing about 320 genera and about 2310 species [1, 2]. Gobioids live in marine, brackish, and freshwater habitats, and form an important component of reef faunas [3–5]. They show a wide range of specializations, including the ability to survive on land for long periods of time (mudskippers; *e.g.* [6]), being diadromous and alternating freshwater and sea water during their lifecycle [7], and forming symbioses with certain shrimp species [8, 9].

Gobioidei belongs to Gobiiformes (*sensu* Thacker et al. [10]), together with Kurtidae, Apogonidae, and Trichonotidae, and consists of nine families: Rhyacichthyidae, Odontobutidae, Milyeringidae, Eleotridae, Butidae, Thalasseleotrididae, Gobiidae, and Oxudercidae (= formerly Gobionellidae [1]), and the extinct †Pirskeniidae [11] (Fig 1). Phylogenetic relationships within extant Gobioidei are generally well resolved based on molecular data [10, 12, 13]. Morphologically, the extant Gobioidei can be divided into two major groups based on the number of branchiostegal rays (bones supporting the gill membrane [14, 15]). Possession of five branchiostegal rays is considered to be the derived state within Gobioidei, supporting the clade Gobiidae + Oxudercidae (denoted 5brG here; see Fig 1), which is also recovered on the basis of molecular data [10, 12, 13]. In contrast, Gobioidei with six branchiostegal rays (6brG) form a paraphyletic group in molecular phylogenies [12, 16], indicating that this condition is plesiomorphic within Gobioidei. The only gobioid that has seven branchiostegal rays, of which the last ray is expanded as is typical for Gobioidei (see [17]), is the genus †*Pirskenius* (denoted 7brG here), and phylogenetic analyses placed it within the paraphyletic 6brG [11]; see Fig 1. The closest living relatives of Gobioidei, the Apogonidae, Kurtidae, and Trichonotidae, all have seven branchiostegals and no expansion of the last branchiostegal ray [1, 17–19].

Previous gobioid classifications date back to more than 100 years ago. Regan [20] had already recognized that two main groups within Gobioidei, which he termed Eleotridae and Gobiidae, can be distinguished based on the configuration of the pelvic fins (separate in Eleotridae vs. united in Gobiidae), the shape of the palatine (L- vs. T-shaped), the endopterygoid (present vs. mostly absent), and the composition of the shoulder girdle (presence vs. absence of the dorsal coracoid [= scapula]). Regan's Eleotridae conforms to the paraphyletic 6brG, and his Gobiidae are reflected in the 5brG. Since then, several efforts have been made to classify groups within Gobioidei using morphological characters (see [21] for a comprehensive compilation). Akihito [22] and Hoese [23] noted that the loss of the anteriormost branchiostegal ray appears to be characteristic for the most advanced groups in Gobioidei (= 5brG). Miller [24] erected the family Rhyacichthyidae and reclassified Gobioidei into Rhyacichthyidae and Gobiidae, with the latter comprising seven subfamilies, Eleotrinae and Xenisthminae (now Eleotridae), Gobionellinae and Tridentigerinae (now included in Oxudercidae), Gobiinae and Kraemeriinae (now Gobiidae), and †Pirskeniinae (referring to †Pirskeniidae Obrhelová, 1961) [10, 12, 25]. Harrison [26] tried to resolve gobioid interrelationships with the help of features of the palatopterygoquadrate complex and indicated some putative relationships among six

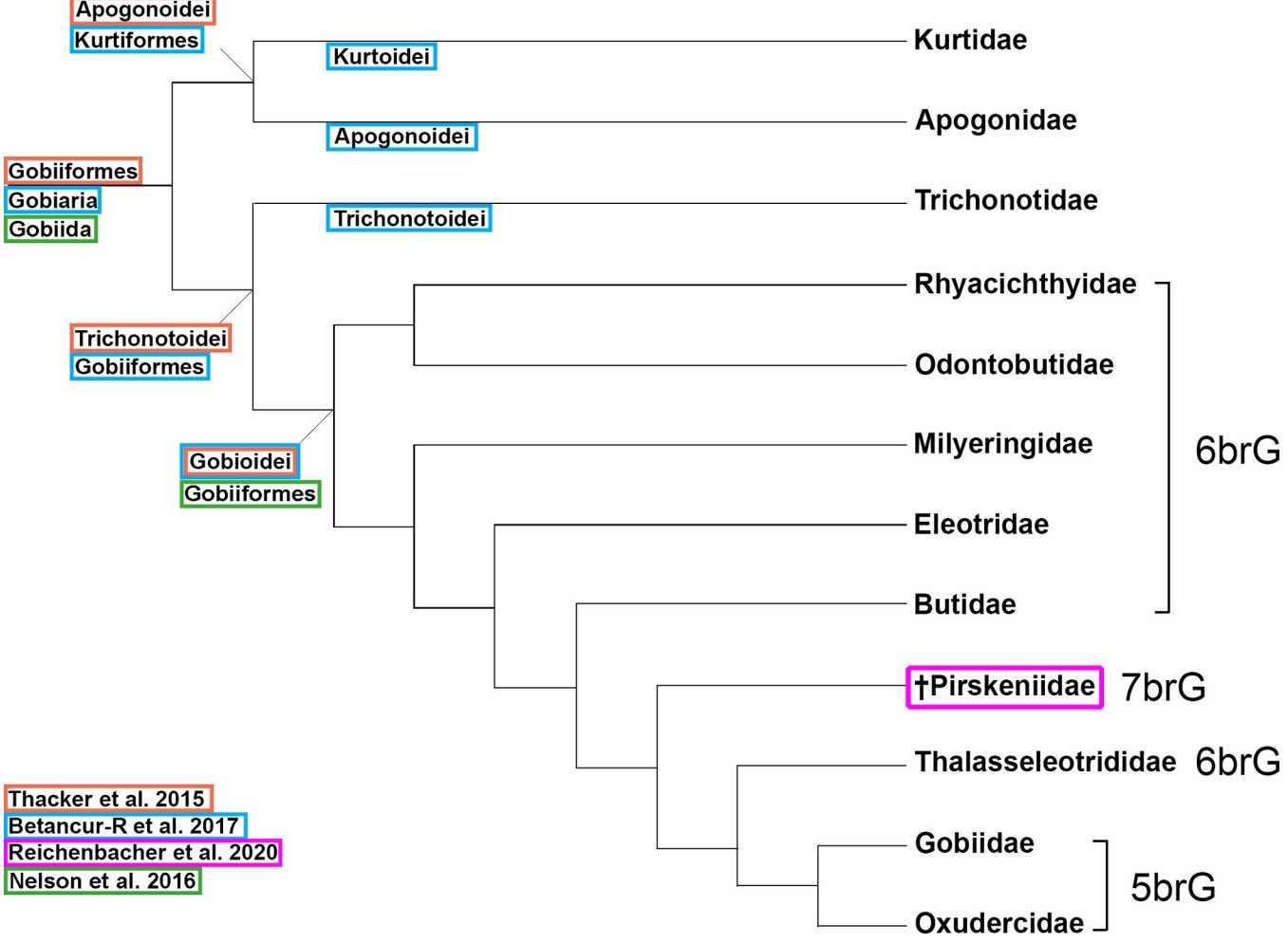

**Fig 1. Current understanding of gobioid relationships showing the different nomenclatures in use.** brG = branchiostegal-rayed Gobioidei.

taxa of the 5brG. Pezold [27] recognized a distinct head pore configuration as an autapomorphy of the subfamily Gobiinae (now in Gobiidae). Hoese and Gill [28] recognized the family Odontobutidae and used 16 characters to propose interrelationships among Rhyacichthyidae, Odontobutidae, and Gobiidae; their Gobiidae included the Butinae, Eleotridinae and Gobiinae, each of which is now assigned family rank [12, 29]. Other authors used morphological data for phylogenetic studies of subgroups of various families, as Larson [30] has done for the *Mugilogobius* group of the Gobiidae, and Murdy [31] for the Oxudercinae. Most of the more recent works that include morphological characters have focused on smaller groups within Gobioidei (e.g. [29, 32–36]).

The known fossil record of Gobioidei is relatively modest in comparison to their recent diversity and comprises about 40 species that are known from skeletal material. In older studies, these species have been assigned to genera or families by comparing meristic characters, such as counts of vertebrae and fin rays (e.g. [37–40]). More recent authors have also considered the systematic value of certain osteological features (e.g. shape of the palatine, first dorsal fin-pterygiophore formula, number of branchiostegal rays, configuration of the palatine-ectopterygoid complex, presence of the entopterygoid), which enabled a tentative systematic placement of some fossil gobioids in relation to extant taxa [41–47].

However, determining the systematic context of fossil gobioids remains a challenge. Firstly, fossil gobioids apparently include several extinct lineages (e.g. [42, 43, 45, 48–51]). Secondly, a phylogenetic analysis of extant gobioids based on morphological characters is hampered by the rarity of synapomorphic characters [28], and becomes even more difficult in fossil gobioids, which usually preserve only skeletal traits and otoliths [11]. Thirdly, all comparative approaches suffer from a limited knowledge of the range of skeletal characters of extant gobioids, which is also due to the sheer number of species (currently 2310, see above) and the inaccessibility of some rare taxa [52, 53]. This explains why, with the exception of †Pirskenii- dae [11], no phylogenetic analyses have yet been conducted to position fossil gobioids within the phylogeny of extant taxa.

The objective of this study was to place ten selected fossil gobioid species within an integra- tive ("total evidence") phylogenetic framework including published molecular sequence data from extant species in combination with morphological data for both fossil and extant species. The choice of fossils was largely based on our previous works (see section Material and Meth- ods). The overall idea was that placement of fossil gobioids in the context of a rigorous phylo- genetic analysis would greatly help to improve our understanding of the evolutionary history of these fascinating fishes.

## Institutional abbreviations

BMNH, former acronym for British Museum of Natural History (now NHMUK); FMNH, Field Museum of Natural History, Chicago, Illinois, USA; IRSNB, Royal Institute of Natural Sciences, Brussel, Belgium; LACM, Los Angeles County Museum of Natural History, Los Angeles, California, USA; MNHN, Muséum national d'Histoire naturelle, Paris, France; MRAC, Musée royal de l'Afrique centrale, Tervuren, Belgium; NHMUK, Natural History Museum in London, United Kingdom; NMP, National Museum, Prague, Czech Republic; SMNS, State Museum for Natural History, Stuttgart, Germany; WAM, Western Australian Museum, Welshpool, Australia; ZM-CBSU, Zoological Museum Collection of the Biology Department at Shiraz University, Iran; ZSM, Bavarian State Collection of Zoology, Munich, Germany.

## Material and methods

### Ethic statement

All specimens originate from museum collections (see Table 1) and no specimens were sacri- ficed for this work.

### Compilation of the taxon set

The ingroup species of the Gobioidei used in this study comprise 29 extant species; the apogo- nid *Sphaeramia nematoptera* is used as outgroup (Table 1). The Odontobutidae, Milyeringi- dae, Butidae, and Thalasseleotridae are each represented by two species, the Rhyacichthyidae and the Eleotridae each by three, the Oxudercidae by five, and the Gobiidae by ten species. Cri- teria for species selection were the availability of 'total-evidence data' including molecular as well as morphological data of the skeleton and otolith data. Morphological data of four species were coded based solely on published information: the odontobutid *Odontobutis obscurus* (data from [44, 54, 55]), the milyeringid *Typhleotris madagascariensis* (data from [56]), the tha- lasseleotridid *Thalasseleotris iota* (data from [57–59]), and the oxudercid *Eucyclogobius new- berryi* (data from [60, 61]); no otolith data were available for *T. madagascariensis*. This selection of extant species, although modest in terms of species numbers of Oxudercidae and

**Table 1. Species and specimens used in this study.** Birdsong et al. [60], Hoese and Gill [28] and Gill and Mooi [15] are used for morphology of many or all species. Colour codes for molecular data: Agorreta et al. [12] = green; Near et al. [63] = blue; Thacker et al. [10] = red; for not color-coded cells see entries in Genbank via given accession numbers.

| Family | Species | Morphology | rRNA | cytb | rag1 | zic1 | sreb2 |
|---|---|---|---|---|---|---|---|
| Fossil | †*Carlomonnius quasigobius* Bannikov and Carnevale, 2016 | Bannikov and Carnevale 2016 [64] | n/a | n/a | n/a | n/a | n/a |
| Fossil | †*Eleogobius brevis* (Agassiz, 1839) | NHMUK PV OR 42779–80; Gierl and Reichenbacher 2015 [42] | n/a | n/a | n/a | n/a | n/a |
| Fossil | †*Eleogobius gaudanti* Gierl and Reichenbacher, 2015 | Gierl and Reichenbacher 2015 [65] | n/a | n/a | n/a | n/a | n/a |
| Fossil | †*"Gobius" francofurtanus* Koken, 1891 | Weiler 1963; Gierl 2012 [62, 66] | n/a | n/a | n/a | n/a | n/a |
| Fossil | †*Gobius jarosi* Prikryl and Reichenbacher, 2018 | Reichenbacher et al. 2018 [46] | n/a | n/a | n/a | n/a | n/a |
| Fossil | †*Lepidocottus aries* (Agassiz, 1839) | Gierl et al. 2013 [44] | n/a | n/a | n/a | n/a | n/a |
| Fossil | †*Paralates bleicheri* Sauvage, 1883 | Gierl and Reichenbacher 2017 [43] | n/a | n/a | n/a | n/a | n/a |
| Fossil | †*Paralates chapelcorneri* Gierl and Reichenbacher, 2017 | Gierl and Reichenbacher 2017 [43] | n/a | n/a | n/a | n/a | n/a |
| Fossil | †*Pirskenius diatomaceus* Obrhelová, 1961 | Obrhelová 1961; Přikryl 2014; Reichenbacher et al. 2020 [11, 49, 67] | n/a | n/a | n/a | n/a | n/a |
| Fossil | †*Pirskenius radoni* Přikryl 2014 | Přikryl 2014; Reichenbacher et al. 2020 [11, 67] | n/a | n/a | n/a | n/a | n/a |
| Gobiidae | *Amblygobius phalaena* (Valenciennes, 1837) | ZSM-PIS-P-GO-0594, -0600; Harrison 1989 [26] | KF415301 | KF415498 | KF415693 | KF415897 | KF416109 |
| Gobiidae | *Aphia minuta* (Risso, 1810) | Rojo 1985; Birdsong et al. 1988; Harrison 1989; La Mesa et al. 2005 (skeleton data) [26, 60, 68, 69]; otolith ex IRSNB | KF415305 | FR851417 | KF415697 | KF415901 | KF416113 |
| Gobiidae | *Asterropteryx semipunctata* Rüppell, 1830 | SMNS-Z-PI-026591, -005882; Birdsong et al. 1988; Van Tassell et al. 1988; Harrison 1989 (skeleton data); Schwarzhans et al. 2020: pl. 7, Fig 5 (otolith) [26, 50, 60, 70] | KF415309 | KF415506 | KF415701 | KF415906 | KF416117 |
| Gobiidae | *Cryptocentrus cinctus* (Herre, 1936) | ZSM-PIS-P-GO-0619, -0639; Karplus and Thompson 2011 [9] | KF415340 | KF415536 | KF415732 | KF415940 | KF416150 |
| Gobiidae | *Discordipinna griessingeri* Hoese and Fourmanoir, 1978 | ZSM-PIS-P-GO-0632, -0633; Hoese and Fourmanoir 1978; Birdsong et al. 1988 [60, 71] | KF415345 | KF415540 | KF415737 | KF415945 | KF416155 |
| Gobiidae | *Glossogobius giuris* (Hamilton, 1822) | ZM-CBSU; Harrison 1989; Esmaeili et al. 2009 [26, 72] | KF415369 | KF415566 | KF415767 | KF415972 | KF416185 |
| Gobiidae | *Gobius niger* Linnaeus, 1758 | NMP6V 146072, - 146073; Harrison 1989 [26] | KF415385 | KF415583 | KF415786 | KF415990 | KF416203 |
| Gobiidae | *Lesueurigobius sanzi* (de Buen, 1918) | ZSM-PIS-035529_1, _4; Miller 1986; Birdsong et al. 1988; Harrison 1989; McKay and Miller 1997 (skeleton data); Schwarzhans et al. 2020: pl. 2, Fig 3 (otolith) [26, 60, 73, 74] | KF415406 | KF415603 | KF415808 | KF416012 | |
| Gobiidae | *Ptereleotris evides* (Jordan and Hubbs, 1925) | ZSM-PIS-P-GO-0587, -0588; Randall and Hoese 1985 [75] | | | KF141341 | KF140623 | KF140263 |
| Gobiidae | *Tigrigobius multifasciatus* (Steindachner, 1876) | ZSM-PIS-P-GO-0622, -0642 | AF491102 | AY846402 | KF415878 | KF416088 | KF416302 |
| Oxudercidae | *Awaous flavus* (Valenciennes, 1837) | ZSM-PIS-43853 (P-GO-1050, -1051) | KF415311 | KF415508 | KF415703 | KF415908 | KF416119 |
| Oxudercidae | *Chlamydogobius eremius* (Zietz, 1896) | ZSM-PIS-43854 (P-GO-1052, -1053); Miller 1987 [76] | KF415329 | KF415526 | KF415720 | KF415928 | KF416138 |
| Oxudercidae | *Eucyclogobius newberryi* (Girard, 1856) | Birdsong et al. 1988; Kindermann et al. 2007 [60, 61]; otolith from the collection of W. Schwarzhans ex LACM-58239 | KF415355 | EU380942 | KF415751 | KF415958 | KF416169 |

*(Continued)*

**Table 1.** (Continued)

| Family | Species | Morphology | rRNA | cytb | rag1 | zic1 | sreb2 |
|---|---|---|---|---|---|---|---|
| Oxudercidae | *Gobioides broussonnetii* Lacepède, 1800 | ZSM-PIS-43852 (P-GO-1048, -1049) | KF415374 | | KF415772 | KF415977 | KF416189 |
| Oxudercidae | *Pomatoschistus flavescens* (Fabricius, 1779) | ZSM-PIS-043982 (2); Vigo, Spain, Sanda, Prag; Mestermann and Zander 1984; Harrison 1989 [26, 77] | KF415386 | KF415584 | KF415787 | KF415991 | KF416204 |
| Thalasseleotrididae | *Grahamichthys* sp. | ZSM-P-GO-0783; Whitley 1956; McDowall 1965; Akihito 1986; Gierl and Reichenbacher 2015 [42, 54, 78–80] | | | KT266412 | KT266520 | KT266467 |
| Thalasseleotrididae | *Thalasseleotris iota* Hoese and Roberts, 2005 | Hoese and Larson 1987; Hoese and Roberts 2005 (skeleton data); Schwarzhans 2019: Fig 99.10 (otolith) [57–59] | | | KT266413 | KT266521 | |
| Butidae | *Kribia nana* (Boulenger, 1901) | MRAC A4-046-P-1116 (1–5); Wongrat 1977 [81] | | AY722211 | KF235468 | | |
| Butidae | *Oxyeleotris marmorata* (Bleeker, 1852) | ZSM-PIS-43857 (P-GO-1058, -1059); Regan 1911; Harrison 1989 [20, 26] | KF415429 | KF415623 | KF415829 | KF416035 | KF416250 |
| Eleotridae | *Dormitator maculatus* (Bloch, 1792) | ZSM-PIS-009383–85; Harrison 1989 (skeleton data) [26]; otolith from the collection of W. Schwarzhans ex LACM (collection Fitch) | KF415347 | KF415542 | KF415739 | KF415947 | KF416157 |
| Eleotridae | *Hypseleotris compressa* (Krefft, 1864) | ZSM-PIS-43863 (P-GO-1068, -1069); Akihito 1986; Akihito et al. 2000; Thacker and Unmack 2005 [14, 25, 54] | KF415398 | KF415596 | KF415800 | KF416004 | KF416217 |
| Eleotridae | *Tateurndina ocellicauda* Nichols, 1955 | ZSM-PIS-43855 (P-GO-1054, -1055) | KF415480 | KF415672 | KF415875 | KF416085 | KF416299 |
| Milyeringidae | *Milyeringa veritas* Whitley, 1945 | WAM-BES1151(B), -BES18735, -BES9768.2; Larson et al. 2013 [82] | | HM590598 | KT266404 | KT266511 | KT266459 |
| Milyeringidae | *Typhleotris madagascariensis* Petit, 1933 | FMNH 116494–98 (skeleton data only); Sparks and Chakrabarty 2012 [56] | | JQ619661 | KY981273 | | |
| Odontobutidae | *Odontobutis obscurus* (Temminck and Schlegel, 1845) | IRSNB ex BMNH 1983.11.4 (otolith only); Iwata et al. 1985; Akihito 1986; Iwata and Sakai 2002; Gierl et al. 2013 [44, 54, 55, 83] | KF415424 | KF415618 | KF415825 | KF416030 | KF416245 |
| Odontobutidae | *Perccottus glenii* Dybowski, 1877 | ZSM-PIS-43867 (P-GO-1078, -1079); Akihito 1986; Birdsong et al. 1988; Harrison 1989 [26, 54, 60] | KF415440 | KF415632 | KF415837 | KF416044 | JX190055 |
| Rhyacichthyidae | *Rhyacichthys aspro* (Valenciennes, 1837) | ZSM-PIS-044262 (BAyFi 13481); Miller 1973; Birdsong et al. 1988 [24, 60] | KF415462 | KF415654 | KF415858 | KF416066 | KF416282 |
| Rhyacichthyidae | *Rhyacichthys guilberti* Dingerkus and Séret, 1992 | MNHN 2019-0113-1, MNHN 2019-0113-2 | | KF669052 | | | |
| Rhyacichthyidae | *Protogobius attiti* Watson and Pöllabauer, 1998 | MNHN 2019–0112; Akihito et al. 2000; Shibukawa et al. 2001 [14, 84] | | AB021257 | | | |
| Apogonidae | *Sphaeramia nematoptera* (Bleeker, 1856) | ZSM-PIS-P-GO-0621, -0634; McAllister 1968; Fraser 1972; Bergman 2004 [19, 85, 86] | AB889673 | | KT266401 | KT266508 | KT266456 |

Gobiidae, was sufficient to achieve a molecular 'backbone' as its phylogenetic analysis produced a well-resolved tree that fully agrees with published hypotheses (see Results).

Ten fossil gobioid species were added to the extant taxon set (Table 1). We selected those that we had examined in previous works [11, 42–44, 46, 62], and added also the oldest known putative gobioid so far, †*Carlomonnius quasigobius* (Table 1). A short overview of all fossil species is provided in S1 Appendix.

## Study and compilation of morphological characters

Literature data, when available, were used to compile phylogenetically informative morphological characters for the extant species (see Table 1 for details). Additionally, X-ray images were

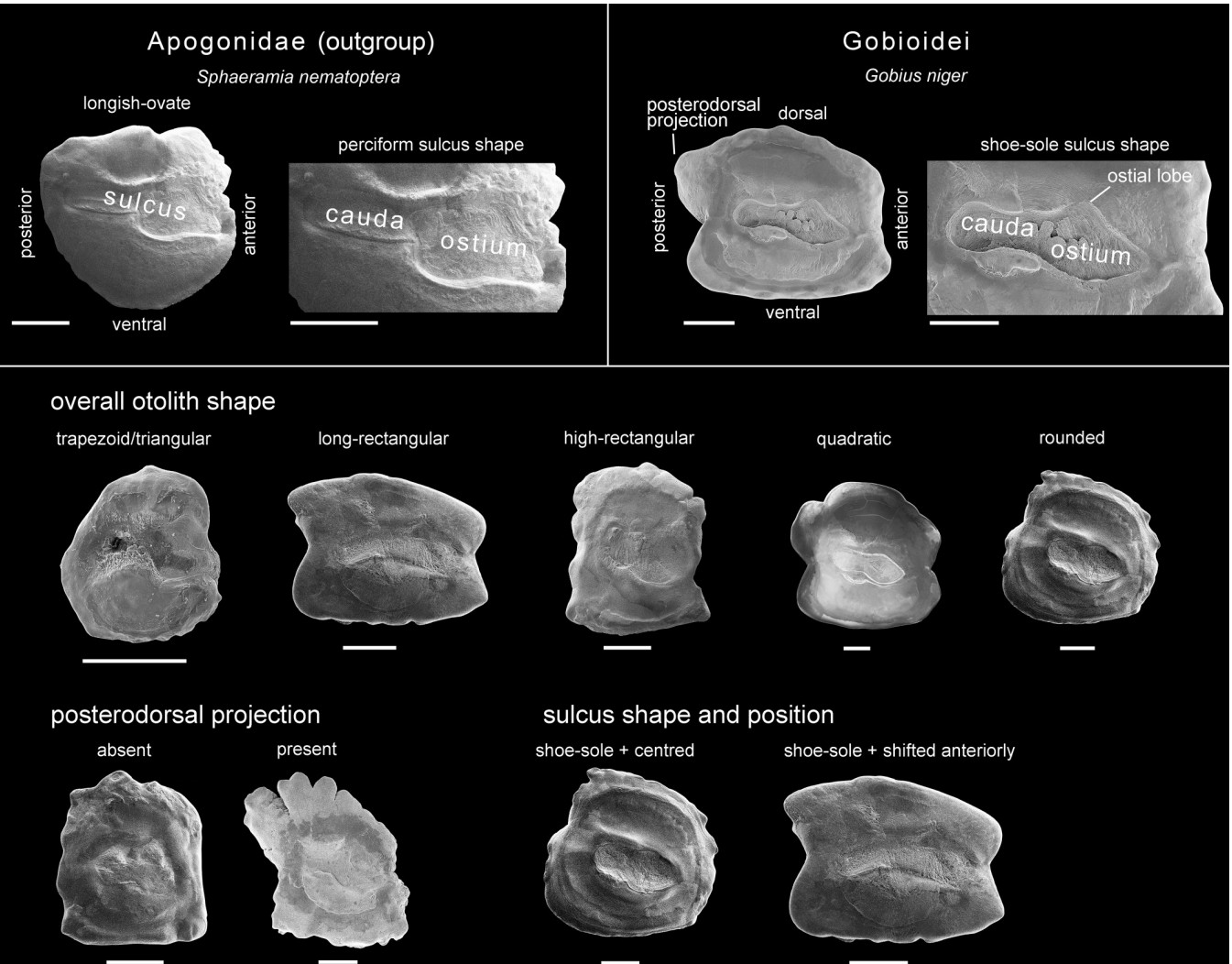

**Fig 2. Otolith terminology and otolith characters and character states used here for the phylogenetic analyses.** Images depict left otoliths (sagittae) in medial view. Scale bars: 0.5 mm.

produced for the extant species with a Faxitron Ultrafocus facility at the ZSM and examined to determine numbers of vertebrae, fin elements and pterygiophores, and configuration of the caudal skeleton. After radiography, otoliths were extracted from the same specimens and prepared for scanning electron microscopy (SEM) imaging (using a HITACHI SU 5000 Schottky FE-SEM at the Department of Earth- and Environmental Sciences, LMU Munich). SEM images served as the basis for the identification of the otolith characters (Fig 2), which are used here for the first time within a phylogenetic matrix. They include (i) the overall otolith shape (six character states: trapezoid/triangular; long rectangular; high rectangular; quadratic; rounded; longish-ovate), (ii) the posterodorsal projection (two states: present; absent), (iii) the sulcus shape (three states: perciform-like, shoe sole, shoe sole/specialized), and (iv) the sulcus shape and position (three states: shoe sole + centred, shoe sole + shifted anteriorly, not shoe sole). Plesiomorphic otolith character states were defined according to the condition seen in the otolith of the outgroup (see S1 Table for details). The otoliths of the included extant gobioid species are shown in Fig 3, the otolith of the outgroup species *Sphaeramia nematoptera*

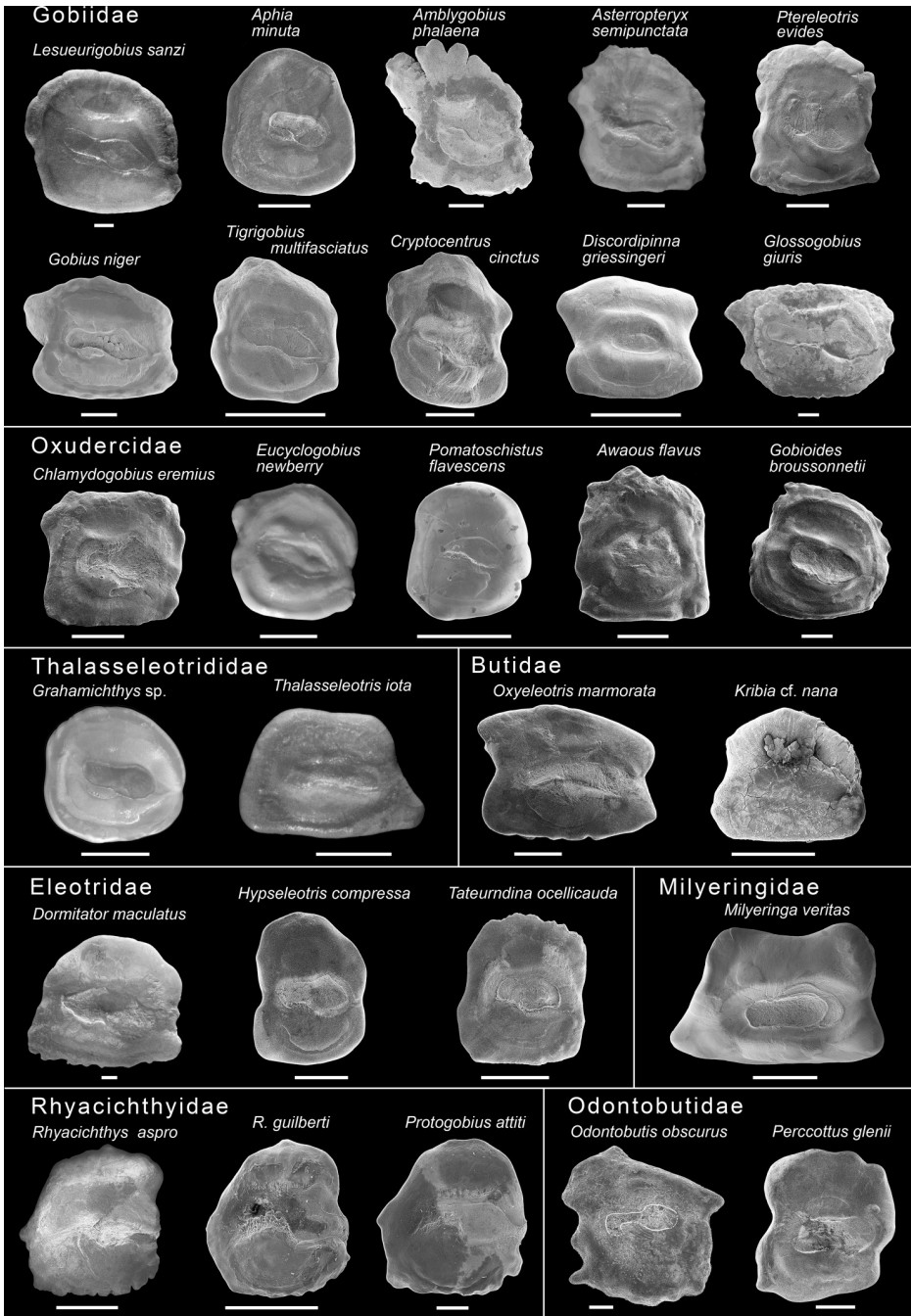

**Fig 3. Otoliths of the extant gobioid species used in this study.** Images depict left otoliths (sagittae) in medial view, except for *Lesueurigobius sanzi*, *Aphia minuta*, *Asterropteryx semipunctata*, *Dormitator maculatus* and *Odontobutis obscurus*, which represent right sagittae that were mirrored for better comparison. Otoliths were not available for the milyeringid *Typhleotris madagascariensis*. For sources of otoliths see Table 1. Scale bars: 0.5 mm.

is depicted in Fig 2. For the fossil species, skeletal and otolith characters were largely compiled from previous works, but some additional characters could also be added (see Results).

A total of 48 morphological characters were assembled. Thirty-eight characters concern bony structures of the skeleton, four characters relate to cartilage, membrane, or tendon

configurations, four refer to otolith morphology, two concern the presence and type of ctenii on the scales, and one is a morphometric character (see S1 Table Part B for all characters). Character states were determined according to literature data and our morphological investigations (based on X-rays, SEM images of otoliths) (see S1 Table Part A for details).

All taxa and characters were assembled in Mesquite 3.61 [87]. We used presence/absence coding (1/0) or up to ten states, depending on the character (see S1 Table).

## Preparation of the molecular data matrix

Molecular sequence data for extant species were assembled based on previously published sequences of five markers: *rDNA* (12S rRNA, tRNA-Val, 16S rRNA), *cytb*, *rag1*, *zic1*, and *sreb2*. The sequences were mainly from the study of Agorreta et al. [12], supplemented by some data from Near et al. [63] and Thacker et al. [10]. Molecular data were downloaded from GenBank, aligned in AliView 1.26 [88] with MUSCLE [89], followed by manual adjustment and exclusion of ambiguous regions where necessary. Individual gene alignments were then concatenated into a supermatrix (6271 bp) with SeaView 5.0.4 [90]. For details and GenBank accession numbers see Table 1. All data matrices are available on figshare (https://figshare.com/s/ed10b9a5ac382f856a20).

## Phylogenetic analyses

We conducted phylogenetic analyses for the extant species based on the morphological character matrix, the molecular supermatrix, and based on the combined molecular and morphological (total evidence) datasets. Adding all ten fossils to the total evidence character set resulted in a collapse of the molecular backbone (see Discussion for possible reasons). Therefore, we used a step-wise approach: (i) a single fossil species was added, (ii) two fossil species of the same genus were added, (iii) four fossil species were added. We used the Bayesian Markov Chain Monte Carlo (MCMC) approach implemented in MrBayes versions 3.2.6 and 3.2.7a [91], as well as implied-weights maximum parsimony (IW-MP, [92]) implemented in TNT 1.5 [93] to infer phylogenies.

In MrBayes, we assigned the Mkv+G model [94, 95] to the morphological data, and separate GTR+G models [95, 96] to each molecular partition. For each analysis, we ran 2 x 4 MCMC chains in parallel for $5 \times 10^6$ generations. We used the "sump" command in MrBayes as well as Tracer 1.7.1 [97] to check for convergence and discarded the first 10% of samples of each analysis as burn-in before summarizing the remaining samples in 50% majority-rule consensus (MRC) trees including posterior probability (PP) values for all clades.

In TNT, we employed new-technology searches (with sectorial search, ratchet, drift, and tree fusing enabled; init. addseqs = 100; find min. length = 10) and a concavity constant of K = 12. A 50% majority-rule consensus tree was calculated if more than one most parsimonious tree was found. For assessing clade support in the IW-MP analysis, we used standard bootstrap resampling [98] (500 replicates; new technology search; init. addseq = 10; find min. length = 5). Phylogenetic trees were visualized in FigTree 1.4.4 [99]; tree files are available on figshare (https://figshare.com/s/ed10b9a5ac382f856a20).

## Results

### Skeleton data

The character state of each character of each species (extant and fossil) is provided in the S1 Table; unknown character states were coded with a question mark.

Two characters could be coded for the first time for †*Eleogobis brevis* based on the specimens NHMUK PV OR 42779 and 42780, deposited in the Natural History Museum in London: the presence of a single anal fin pterygiophore inserting before the haemal spine of the first caudal vertebra (AP = 1), and the position of the penultimate branchiostegal on the ceratohyal. Re-inspection of the type specimens of †*Gobius jarosi* revealed that also in this species the penultimate branchiostegal is located on the ceratohyal. Finally, a count of three to four anal fin pterygiophores (AP = 3–4) could be determined for †*Lepidocottus aries* based on re-inspection of the specimens used in Gierl et al. [44].

## Otolith data

Fig 3 depicts the otoliths of the extant species included in this study. Short descriptions of the otoliths are given below.

**Family Gobiidae.** *Amblygobius phalaena (Valenciennes, 1837).*–High-rectangular, margins crenate, dorsal rim strongly lobed, pronounced posterodorsal projection present. Sulcus centered, shoe-sole shape with slender cauda and wide ostium with dorsal lobe.

*Aphia minuta (Risso, 1810).*–Rounded, margins smooth, short projection slightly below level of ostium tip. Sulcus shifted anteriorly, rather shoe-sole shape, cauda strongly reduced, ostium with dorsal lobe and pointed tip.

*Asterropteryx semipunctata Rüppell, 1830.*–Rectangular, higher than long. Dorsal, anterior and ventral margins slightly crenulate, small posterodorsal projection. Sulcus centered, shoe-sole shape, with rounded cauda and ostium. See Schwarzhans et al. [50] for additional details of the same otolith.

*Cryptocentrus cinctus (Herre, 1936).*–High-rectangular, margins smooth, with a projection in the middle of the dorsal margin and a median constriction on anterior and posterior rims. Sulcus centered, shoe-sole shape with small, rounded cauda and a broad ostium with ostial lobe.

*Discordipinna griessingeri Hoese and Fourmanoir, 1978.*–Long-rectangular, smooth margins with median constrictions anterior and posterior defining four projections. Sulcus centered, shortened, oval specialized, surrounded by a bulging crista.

*Glossogobius giuris (Hamilton, 1822).*–Long-rectangular, slightly crenate margins, posterodorsal projection and a sub-median projection on the posterior rim. Sulcus centered, shoe-sole shape, cauda elongated, ostium with strong dorsal lobe.

*Gobius niger Linnaeus, 1758.*–Long-rectangular, dorsal margin slightly undulated, prominent posterodorsal and posteroventral projections, small anteroventral projection. Sulcus centered, shoe-sole shape, elongated with rounded cauda and ostium with dorsal lobe.

*Lesueurigobius sanzi (de Buen 1918).*–Quadratic with rounded dorsal margin, dorsal and anterior margins faintly lobed, other margins smooth, submedian constriction on posterior margin. Posterodorsal projection short and rounded, weak anteroventral projection. Sulcus centered, shoe-sole shape, rounded cauda, ostium with pointed tip. See Schwarzhans et al. [50] for additional details of the same otolith.

*Ptereleotris evides (Jordan and Hubbs, 1925).*–High-rectangular, margins sinuate; small posteroventral and more pronounced anteroventral projection. Sulcus centered, shoe-sole shape, with rounded cauda and ostium.

*Tigrigobius multifasciatus (Steindachner, 1876).*–Quadratic, margins undulated, cusp on dorsal rim. Sulcus shifted anteriorly, shoe-sole shape, cauda rounded, ostium with pronounced elongate tip.

**Family Oxudercidae.** *Awaous flavus (Valenciennes, 1837).*–High-rectangular, dorsal and posterior margins crenate. Sulcus centered, shoe-sole shape. Cauda and ostium rounded, ostium with marked dorsal lobe.

*Chlamydogobius eremius (Zietz, 1896).*–Quadratic, margins slightly undulated, small posterodorsal projection. Sulcus centered, shoe-sole shape, with small, rounded cauda and broad ostium with marked dorsal lobe.

*Eucyclogobius newberryi (Girard, 1856).*–Quadratic with slightly curved dorsal margin, small posterodorsal projection, sulcus centered, shoe-sole shape, with well-developed cauda and ostium.

*Gobioides broussonnetii Lacepède, 1800.*–Rounded, margins irregularly crenate, small posterodorsal cusp. Sulcus centered, shoe-sole shape, rounded cauda, large ostium with rounded tip.

*Pomatoschistus flavescens (Fabricius, 1779).*–Quadratic, margins smooth. Sulcus centered, shoe-sole shape, weakly developed cauda, rounded ostium.

**Family Thalasseleotrididae.** *Grahamichthys sp.*–Rounded, all margins smooth, no projections. Sulcus centered, shoe-sole shape, cauda and ostium rounded.

*Thalasseleotris iota Hoese and Roberts, 2005.*–Long-rectangular, margins smooth, with prominent praeventral projection and slightly rounded anterodorsal expansion. Sulcus centered, shoe-sole shape, cauda and ostium slender. See Schwarzhans [58] for additional details of the same otolith.

**Family Butidae.** *Kribia cf. nana (Boulenger, 1901).*–Long-rectangular, all margins relatively smooth, small posterodorsal projection, posteroventral expansion (bulge). Sulcus centered, shoe-sole shape, slightly shifted anteriorly. Rounded cauda about half the length of the ostium.

*Oxyeleotris marmorata (Bleeker, 1852).*–Long-rectangular, anterior and posterior margins distinctly incised in the middle, pointed posterodorsal and rounded anterodorsal projections. Sulcus centered, slender-to-shoe-sole shape, slightly shifted anteriorly.

**Family Eleotridae.** *Dormitator maculatus (Bloch, 1792).*–Long-rectangular, dorsally rounded. Anterior and ventral margins finely crenulated, other margins smooth. Sulcus centered, shoe-sole shape; cauda almost as large as ostium and posteriorly pointed.

*Hypseleotris compressa (Krefft, 1864).*–High-rectangular. All margins smooth, anterior and posterior margins with sub-median constriction at the level of the sulcus. Sulcus centered, shoe-sole shape, rounded cauda. Ostium with angled dorsal lobe.

*Tateurndina ocellicauda Nichols, 1955.*–High-rectangular, all margins relatively smooth except for dorsal margin, which is serrate. Sulcus centered, rounded-to-shoe-sole shape, cauda and ostium only weakly differentiated.

**Family Milyeringidae.** *Milyeringa veritas Whitley, 1945.*–Trapezoid and smooth with posteroventral projection and dorsally pointing anterodorsal projection. Sulcus centered, rounded-to-shoe-sole shape, surrounded by a crista, cauda and ostium not clearly differentiated.

**Family Odontobutidae.** *Odontobutis obscurus (Temminck and Schlegel, 1845).*–Trapezoid, dorsal margin with tip in its posterior half, anterior margin with spine-like dorsal projection, ventral margin crenulated and with spine-like posterior projection, posterior margin deeply sinuate. Sulcus centered, supra-median, relatively small, perciform-like. Cauda slender, posteriorly bent, ostium rounded and distant from anterior rim.

*Perccottus glenii Dybowski, 1877.*–Trapezoid with sinuate dorsal and ventral margins, posterior margin with median notch, anterior margin with sub-median notch. Sulcus centered, rounded-to-shoe-sole shape, cauda slightly shorter than ostium.

**Family Rhyacichthyidae.** *Protogobius attiti Watson and Pöllabauer, 1998.*–Trapezoid/triangular, rounded with short median projection on posterior margin, dorsal margin smooth, ventral margin lobed. Sulcus shifted anteriorly, perciform-like with slender, relatively short cauda and broad ostium; ostium opened to anterior rim.

*Rhyacichthys aspro (Valenciennes, 1837)*.–Trapezoid/triangular with short median projection on posterior and dorsal margins, dorsal margin slightly undulated with prominent antero-dorsal bulge, ventral margin crenate. Sulcus shifted anteriorly, perciform-like with slender, relatively short cauda and broad ostium; ostium anteriorly closed.

*Rhyacichthys guilberti Dingerkus and Séret, 1992*.–Trapezoid/triangular, ventral margin lobed, dorsal margin smooth. Sulcus shifted anteriorly, perciform-like with slender, relatively short cauda and broad ostium; ostium opened to anterior rim.

**Family Apogonidae (outgroup).** *Sphaeramia nematoptera (Bleeker, 1856)*.–Overall shape longish-ovate, with projections on the shallower dorsal rim. Perciform sulcus slightly above the middle, with broad ostium and slender cauda which are of equal length.

## Phylogenetic analyses

**Phylogenetic relationships of the extant species.** *Molecular data*.–The Bayesian phylogeny based on molecular data (Fig 4A) recovers all families as monophyletic. Notably, although we used a restricted number of species, the tree is completely congruent with the trees published by Agorreta et al. [12] and Thacker et al. [10]: Rhyacichthyidae and Odontobutidae are sister groups, and together they are sister to the rest of the gobioid families. Milyeringidae, Eleotridae, Butidae, and Thalasseleotrididae are successive sister groups to the 5brG clade, which is composed of well-supported Oxudercidae and Gobiidae.

Maximum Parsimony analysis produced a single most parsimonious tree (S1 Fig in S1 File), which is similar to the Bayesian tree, but some nodes have poor bootstrap support (BS). The tree is topologically identical to the Bayesian tree on family-level, only within Gobiidae there are some minor differences concerning the positions of *Amblygobius* and *Asterropteryx*; BS for several nodes is rather low compared to the support in the Bayesian tree.

*Morphological data*.–In the Bayesian phylogeny restricted to the extant species, the 5brG clade (Gobiidae + Oxudercidae) is recovered with maximum support, but the internal structure of the clade is completely unresolved (Fig 4B). The thalasseleotridid species *G. radiatus* and *Th. iota* group together and are sister to 5brG. The Thalasseleotrididae + 5brG clade is maximally supported, consistent with molecular phylogenetic results. Also, similar to the established molecular phylogeny, Eleotridae, Butidae, and Milyeringidae are closely related to that clade (1.00). However, only Eleotridae and Milyeringidae are recovered as monophyletic (0.91, 0.82), whereas Butidae are paraphyletic (Fig 4B). Monophyly of Odontobutidae is also not resolved, and in contrast to the molecular phylogeny the odontobutid species are closer to the above assemblage (1.00) instead of being sister to Rhyacichthyidae, which are recovered paraphyletic (Fig 4B).

The Maximum Parsimony analysis recovered one most parsimonious tree (S2 Fig in S1 File), which is overall similar to the Bayesian tree, but many nodes have very poor bootstrap support (BS). Within the 5brG clade (BS = 95%) the reciprocal monophyly of Gobiidae and Oxudercidae is not recovered. Thalasseleotrididae is monophyletic and highly supported as sister to 5brG (99%). As in the Bayesian tree, among the remaining families Eleotridae (71%) and Milyeringidae (52%) are supported as monophyletic.

*Total-evidence approach*.–The Bayesian phylogeny inferred from the total-evidence dataset (= combined molecular and morphological data) including only extant species is topologically identical to the molecular phylogeny but shows maximum support for Rhyacichthyidae and Rhyacichthyidae + Odontobutidae (1.00 and 1.00 vs. 0.62 and 0.64, respectively) (S3 Fig in S1 File).

The Maximum Parsimony analysis recovered one most parsimonious tree (S4 Fig in S1 File), which is similar to the Bayesian tree. Most nodes have good bootstrap support, except for

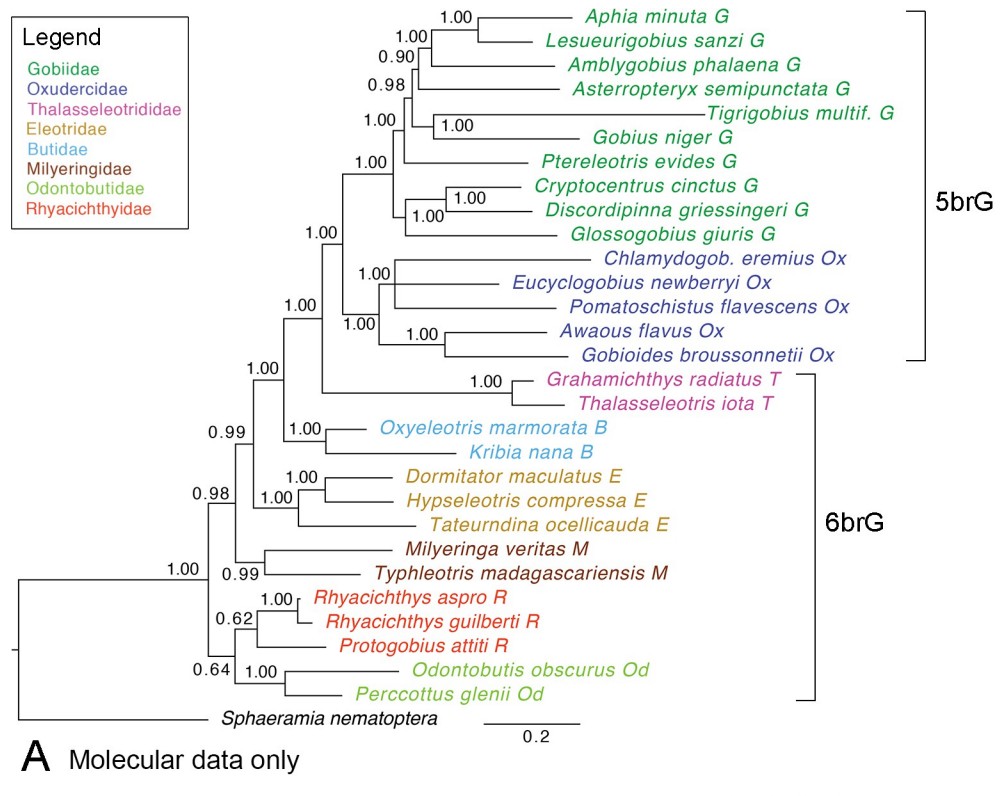

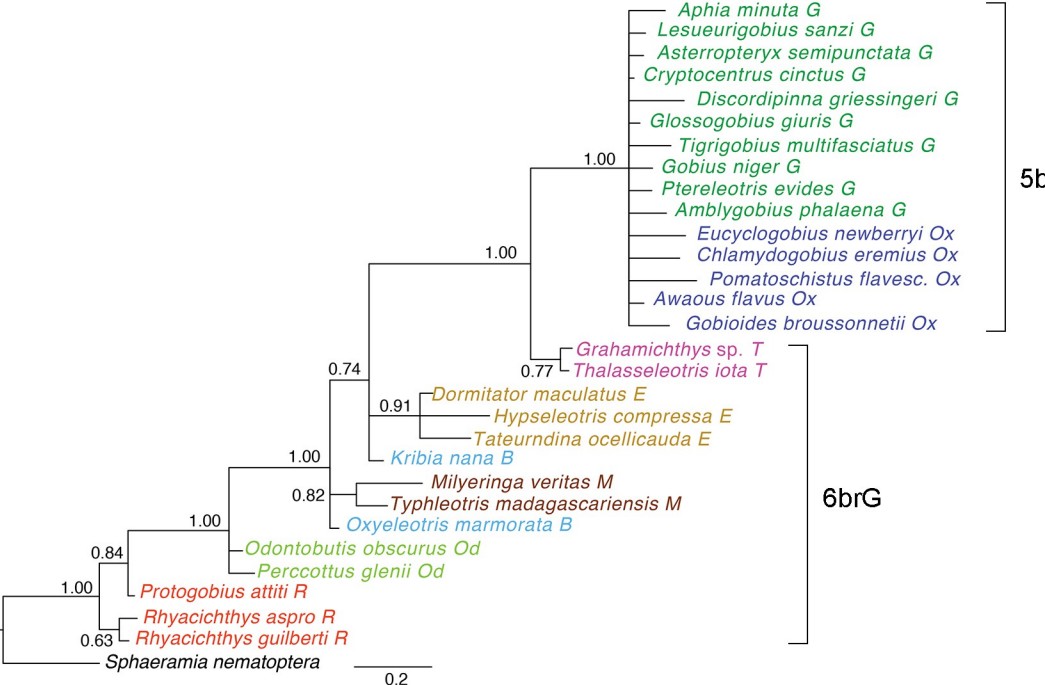

**Fig 4. 50% MRC Bayesian tree with posterior probabilities. A** Tree based on published DNA data of the 29 gobioid species used in this study (average standard deviation of split frequencies between two independent runs [ASDSF] = 0.003776; for sources of molecular data see Table 1). **B** Tree based on morphological characters of the extant species only (ASDSF = 0.002446). Scale bar, average number of substitutions per site respectively character changes per character.

some clades within 5brG that are weakly supported. Concerning the topology, the only difference is that the positions of the oxudercid genera *Eucyclogobius*, *Pomatoschistus* and *Chlamydogobius* are resolved while they form a polytomy in the Bayesian tree (see S3 Fig vs. S4 Fig in S1 File).

**Phylogenies including extant and all ten fossil species.** *Morphological data.*–In the Bayesian phylogeny, the 5brG clade (Gobiidae + Oxudercidae)–including the two fossil *Gobius* spp.–is recovered only with moderate support (0.70), and internal relationships are again unresolved (Fig 5A). The rest of the tree is topologically similar to the one based on the morphology of the extant species only (Fig 4B), Thalasseleotrididae are now recovered with slightly higher support (0.80 vs. 0.77). The two †*Eleogobius* spp. are placed between Thalasseleotrididae and 5brG (0.60), but monophyly of the genus is not resolved. The †*Pirskenius* and †*Paralates* spp. form a weakly supported clade (0.64) that is sister to the above assemblage (0.73), but only †*Pirskenius* (i.e., †Pirskeniidae) is resolved as monophyletic (0.87). The remaining two fossil species, †*Lepidocottus aries* and †*Carlomonnius quasigobius*, are clearly more related to Thalasseleotrididae + 5brG than to Rhyacichthyidae and Odontobutidae (0.84), but their exact placements are not resolved.

The Maximum Parsimony analysis using the same data set recovered six most parsimonious trees. The resulting 50% majority rule consensus tree (S5 Fig in S1 File) is overall consistent with the one based on extant species only (S2 Fig in S1 File) but is very poorly resolved on a deeper level; most nodes have BS < 50%. The two fossil *Gobius* spp. are included in the 5brG in a clade with *Gobius niger*, *Discordipinna* and *Lesueurigobius*. The two †*Eleogobius* spp. are sister to 5brG but monophyly of the genus remains unresolved. †*Carlomonnius quasigobius* is placed in a clade containing Eleotridae and *Kribia nana* (Butidae). †*Pirskenius* is monophyletic (64%) but its position, as well as that of the remaining three fossil species, is not further resolved.

*Total-evidence approach.*–The Bayesian phylogenetic analysis of the total-evidence dataset including all 29 extant and the ten fossil species produced a consensus tree with a largely collapsed backbone (Fig 5B) compared to the tree based on only the morphological data set of the same taxa (Fig 5A). It contains numerous polytomies and shows poor support for many deeper nodes. The 5brG clade (Gobiidae + Oxudercidae) forms a polytomy with the Thalasseleotrididae, the Butidae, the clade of †*Paralates* + †*Pirskenius* and three further fossil species (†*Eleogobius* brevis, †*E. gaudanti*, †"*Gobius*" *francofurtanus*). Nevertheless, several groups can be recovered within this polytomy. Oxudercidae, which were not resolved in the phylogeny based only on morphology (Fig 5A), are now recovered with high support (0.93), as well as some clades within Gobiidae that correspond to the molecular phylogeny, i.e. the *Aphia* (0.89) and *Glossogobius* (0.72) clades. †*Gobius jarosi* is weakly (0.55) placed as sister to the extant *G. niger*. The clade of †*Pirskenius* + †*Paralates* (see above) is recovered with slightly weaker support (0.59 vs. 0.64), while support for monophyly of †Pirskeniidae is slightly increased (0.93 vs. 0.87). †*Lepidocottus aries* and †*Carlomonnius quasigobius* are resolved as members of Butidae, albeit with low support (0.61).

The consensus tree of the two most parsimonious trees of the same data set (S6 Fig in S1 File) is even less well resolved, with very weak BS for many nodes. The 5brG clade is recovered, with reciprocally monophyletic Oxudercidae and Gobiidae. In contrast to the Bayesian tree, †"*Gobius*" *francofurtanus* is sister to *Gobius niger* (< 50%) (vs. not resolved in the Bayesian tree), whereas †*Gobius jarosi* is sister to *Lesueurigobius* (also < 50%) (vs. sister to *G. niger*). †*Eleogobius* is monophyletic and sister to the 5brG clade, but again BS for this is negligible. †*Pirskenius* is monophyletic (65%) but its position, as well as that of the remaining four fossil species, is not further resolved.

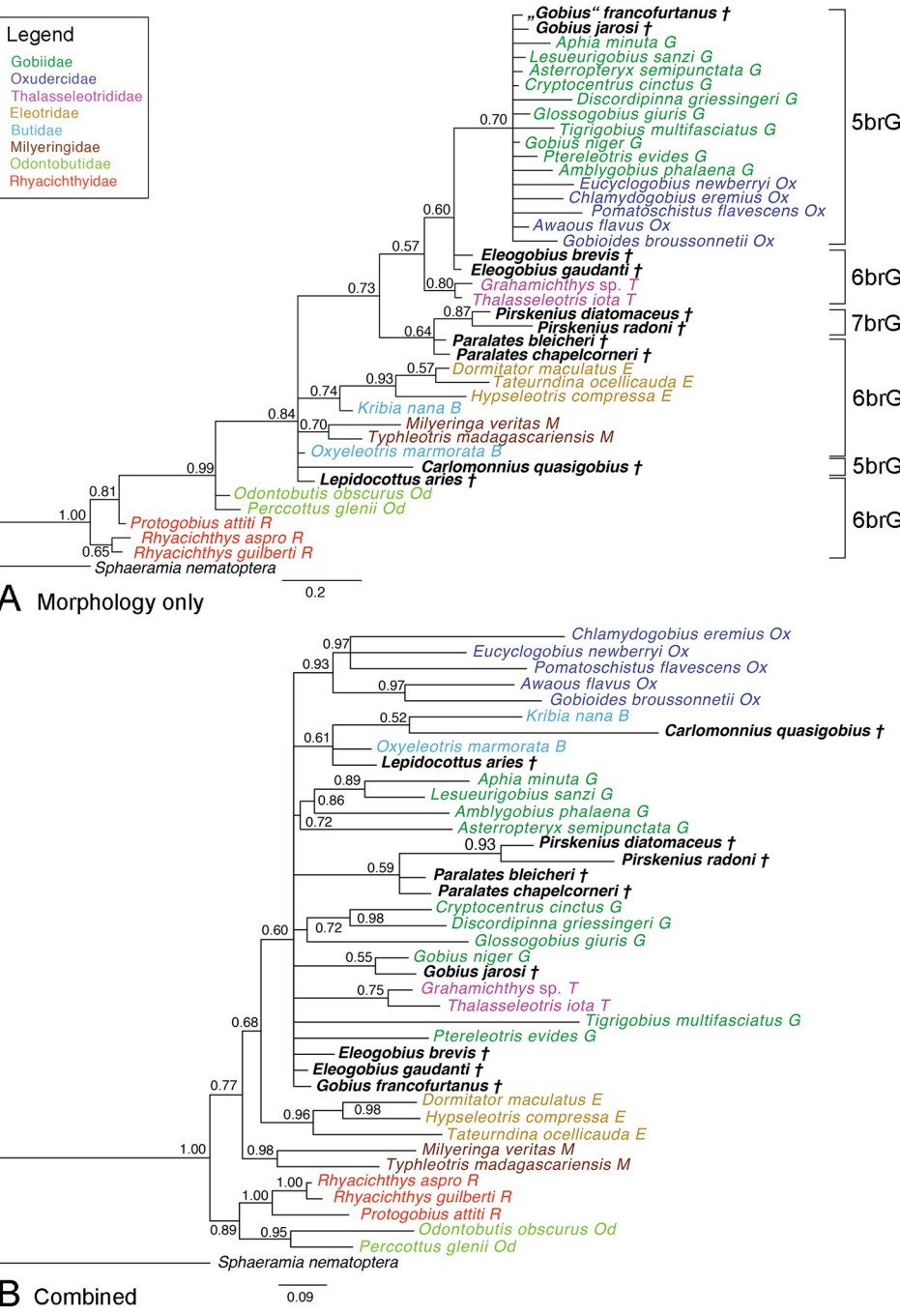

**Fig 5. 50% MRC Bayesian trees with posterior probabilities based on the extant plus ten fossil species. A** Tree based on only the morphological data set (ASDSF = 0.006790). **B** Tree based on the total evidence data set (ASDSF = 0.010856). Scale bars, average number of substitutions per site respectively character changes per character.

**Total-evidence phylogenies including one to four fossil species.** In the following analyses, a single fossil was added to the data set of the extant species and the trees were inferred based on the morphological and molecular data (total-evidence approach).

†*Carlomonnius quasigobius.*–The Bayesian phylogeny is topologically identical to the molecular and the total evidence phylogenies based on extant species only; †*Carlomonnius*

*quasigobius* is placed within Butidae (Fig 6A). Support values are high (>0.90) for most clades, but not for Butidae (0.74 vs. 1.00 in the molecular tree). The single Maximum Parsimony tree retained recovered all recent families, like the Bayesian tree (S7 Fig in S1 File). However, in this tree †*C. quasigobius* is placed as sister to Thalasseleotrididae + 5brG, albeit with very low support (< 50%).

†*Lepidocottus aries*.–The Bayesian phylogeny (Fig 6B) and the single Maximum Parsimony tree retained (S8 Fig in S1 File) reveal the same topology and almost the same support values as described above for the trees including †*C. quasigobius*. †*Lepidocottus aries* is placed within Butidae in both the Bayesian and the Maximum Parsimony tree, with high support values in the former (0.95), but very low support in the latter (< 50%).

†*Gobius jarosi and* †*"Gobius" francofurtanus*.–In the Bayesian tree (Fig 7A), topologies and support values are similar as described above. In comparison to the Bayesian total-evidence phylogeny based on extant taxa only (S3 Fig in S1 File), somewhat decreased support values concern two internal gobiid clades: the *Aphia*-lineage (0.95 vs. 1.00) and the clade containing *Glossogobius* and two members of the *Cryptocentrus*-lineage (0.90 vs. 1.00). The two fossil species are recovered as successive sister groups to *G. niger* with moderate support (0.77) (Fig 5D). In the consensus tree of two Maximum Parsimony trees (S9 Fig in S1 File), †*"Gobius" francofurtanus* is sister to *Gobius niger* while the position of †*G. jarosi* is unresolved in a clade with *Lesueurigobius sanzi*, *Tigrigobius multifasciatus*, *Asterropteryx semipunctata*, *Amblygobius phalaena*, and *Aphia minuta* (S9 Fig in S1 File). When only one of the two fossil species is included, a sister-relation with *G. niger* is apparent in each case, in both the Bayesian and the Maximum Parsimony tree (S10–S13 Figs in S1 File).

†*Eleogobius brevis and* †*E. gaudanti*.–The Bayesian phylogeny inferred from the total-evidence dataset including all extant species and either †*E. brevis* or †*E. gaudanti* (S14, S15 Figs in S1 File) is topologically identical to the molecular phylogeny, and almost no decrease of support values is seen. †*Eleogobius brevis* is recovered as sister to *Gobius niger* with good support (0.87), while †*E. gaudanti* shows a well-supported (0.87) sister relation to the Thalasseleotrididae. When both species of †*Eleogobius* are added, they are recovered in a polytomy with the Thalasseleotrididae (0.88) (Fig 7B), and there is slightly decreased support for the 5brG clade (0.90 vs. 1.00), and also for some of the gobiid clades (e.g. *Tigrigobius* + *G. niger*: 0.91 vs. 1.00) compared to the total evidence tree using only the extant species (S3 Fig in S1 File).

In the Maximum Parsimony trees, when only one of the †*Eleogobius* spp. is included, its position matches that recovered in the Bayesian analyses, albeit with poor support (S16, S17 Figs in S1 File). When both species are included, they form a clade and are sister to the Thalasseleotrididae (< 50%, S18 Fig in S1 File).

†*Pirskenius diatomaceus and* †*P. radoni*.–In the Bayesian tree, topologies are the same as described above; the two species of †*Pirskenius* are recovered as sister to Thalasseleotrididae (Fig 8A). However, in comparison to the Bayesian total-evidence phylogeny based on extant taxa only, decreased support values occur for the 5brG clade (0.89 vs. 1.00), the Thalasseleotrididae (0.65 vs. 1.00), and for the gobiid clade of *Tigrigobius* + *Gobius niger* (0.90 vs. 1.00). Support for the clade Thalasseleotrididae plus one of the †*Pirskenius* species is similar when only †*P. radoni* is added (0.69) (S19 Fig in S1 File), but higher when only †*P. diatomaceus* is involved (0.91) (S20 Fig in S1 File).

The single Maximum Parsimony tree including both species of †*Pirskenius* recovers all recent families as monophyletic (S21 Fig in S1 File) and the genus is recovered as sister to Thalasseleotrididae + 5brG; its monophyly is supported with 74% BS. Adding solely †*P. radoni* or †*P. diatomaceus* results in the same topology and similar support values (S22, S23 Figs in S1 File) as seen in the tree including both species.

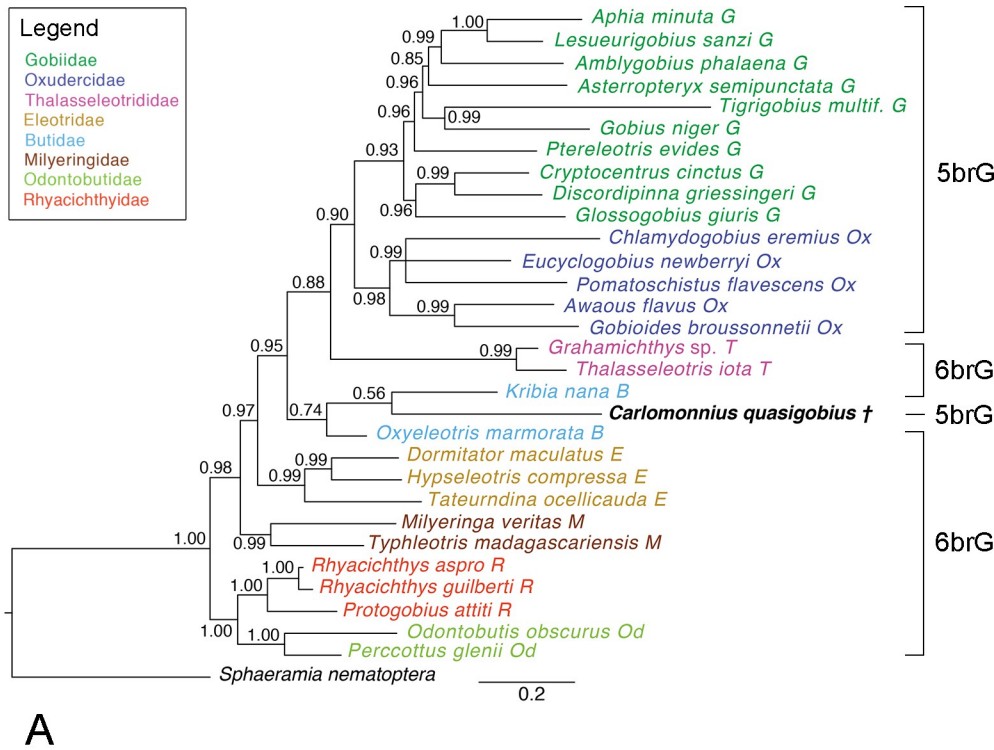

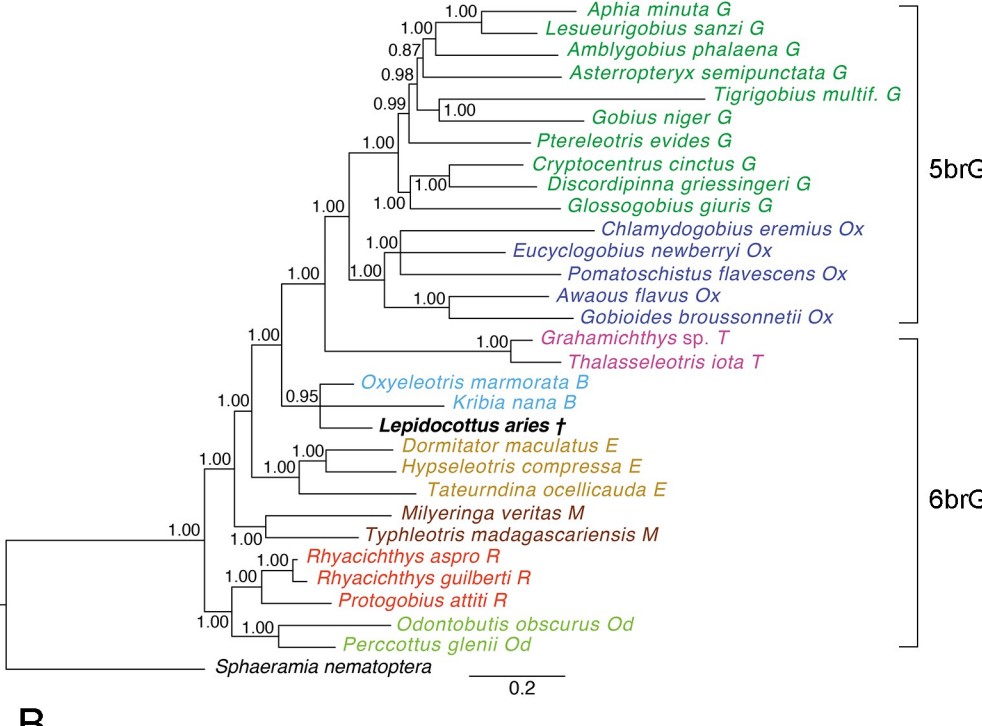

**Fig 6. 50% MRC Bayesian trees with posterior probabilities based on the total-evidence dataset. A** †*Carlomonnius quasigobius* was added to the extant species (ASDSF = 0.003939). **B** †*Lepidocottus aries* was added to the extant species (ASDSF = 0.000854). Scale bars, average number of substitutions per site respectively character changes per character.

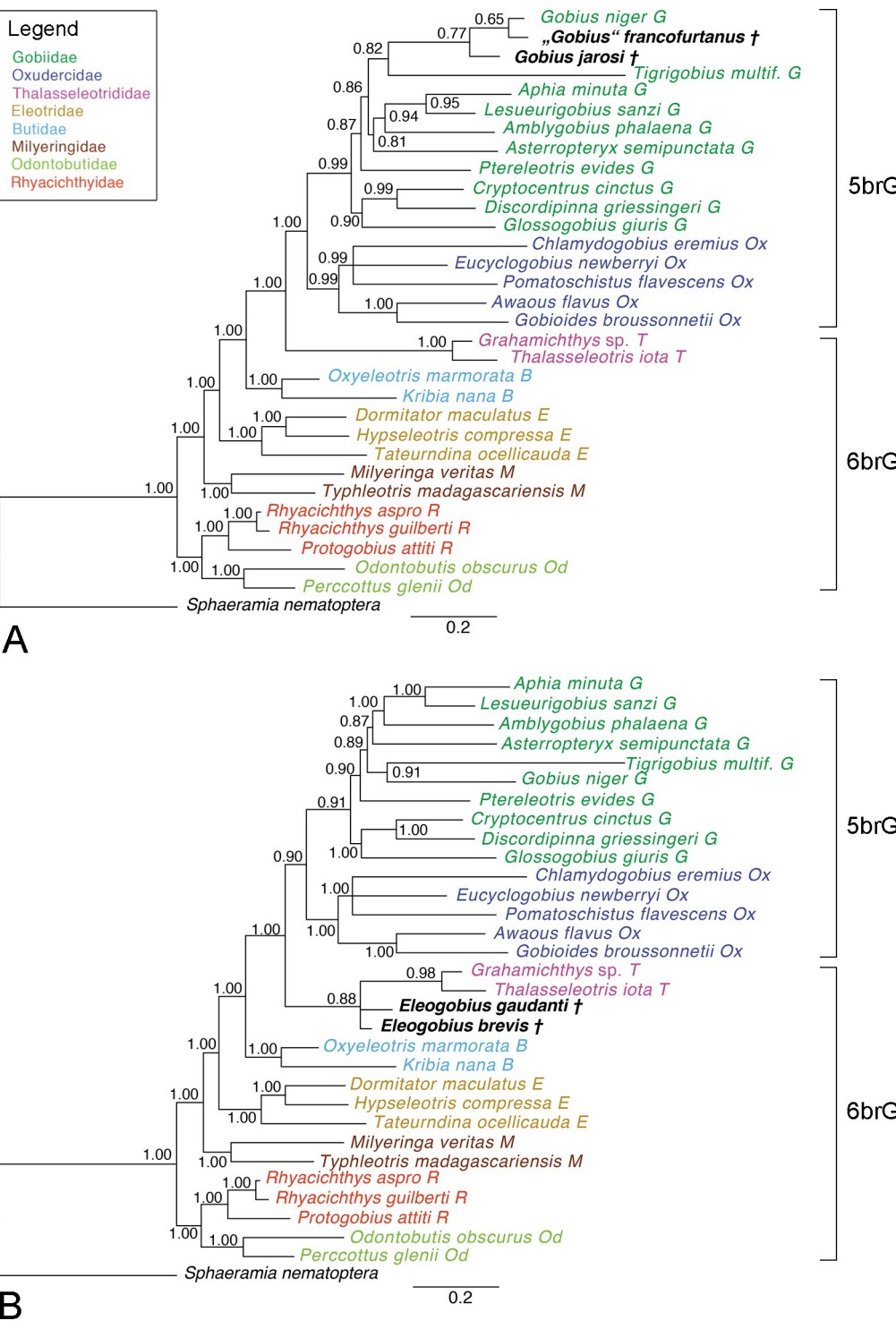

**Fig 7. 50% MRC Bayesian trees with posterior probabilities based on the total-evidence dataset. A** †"*Gobius*" *francofurtanus* and †*G. jarosi* were added to the extant species (ASDSF = 0.014107). **B** Both †*Eleogobius* species were added to the extant species (ASDSF = 0.004114). Scale bars, average number of substitutions per site respectively character changes per character.

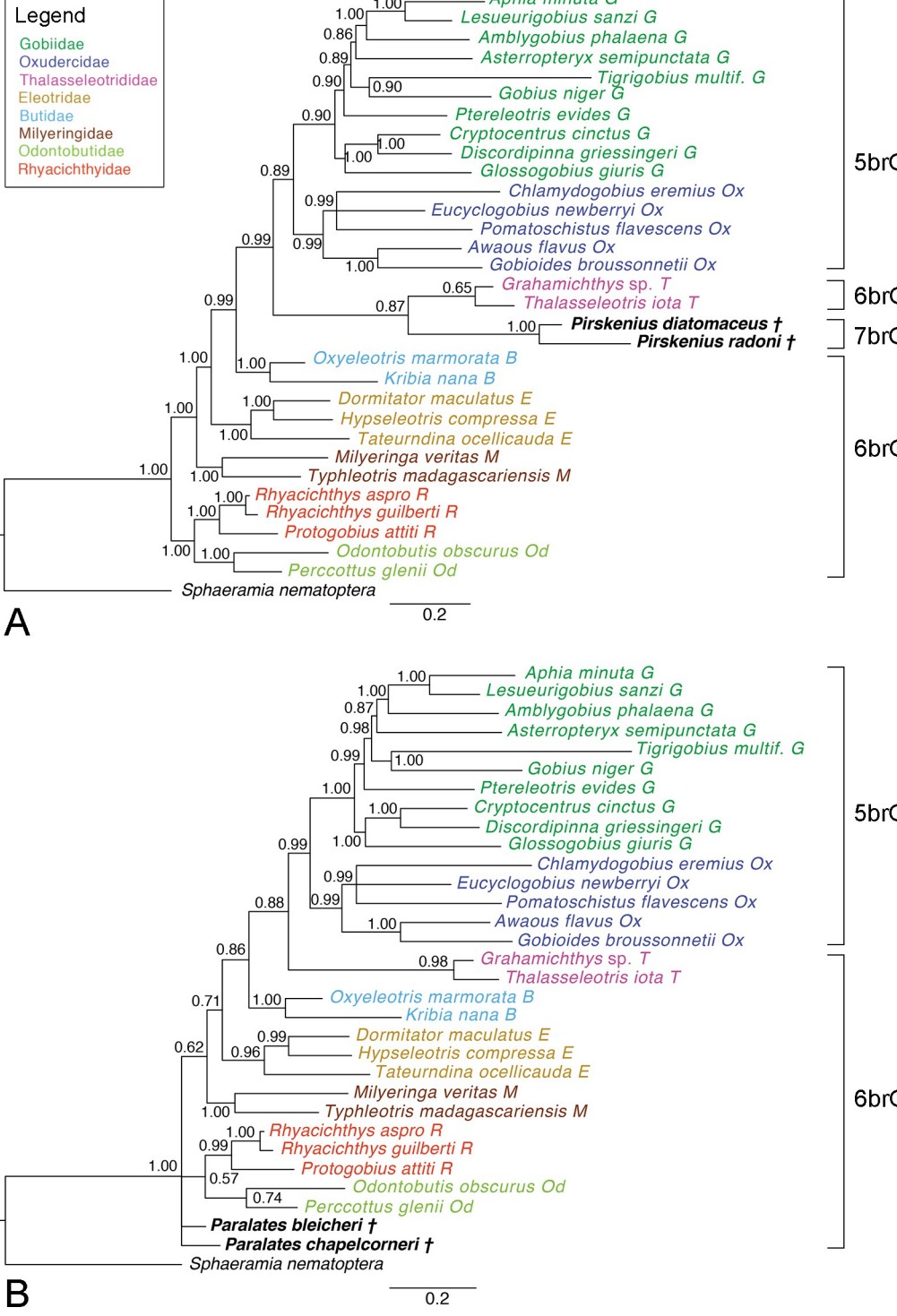

**Fig 8. 50% MRC Bayesian trees with posterior probabilities based on the total-evidence dataset. A** †*Pirskenius diatomaceus* and †*P. radoni* were added to the extant species (ASDSF = 0.017673). **B** †*Paralates bleicheri* and †*Pa. chapelcorneri* were added to the extant species (ASDSF = 0.006995). Scale bars, average number of substitutions per site respectively character changes per character.

†*Paralates bleicheri and* †*Pa. chapelcorneri.*–Inclusion of both species of †*Paralates* does not recover a relationship of these two fossil taxa with any of the extant clades in the Bayesian phylogeny, rather they form a polytomy with the Odontobutidae + Rhyacichthyidae clade and the clade containing all other families (Fig 8B). The Bayesian phylogeny that includes only †*Pa. bleicheri* resolves this species as sister to Odontobutidae (S24 Fig in S1 File), albeit with relatively low support (0.60). In comparison with the total-evidence phylogeny including only extant taxa, support for Odontobutidae decreased (0.61 vs. 1.00), while the high support for all other deeper nodes was not affected. The Bayesian phylogeny that contains only †*Pa. chapelcorneri* recovers this species as sister to the clade containing the 5brG, Thalasseleotrididae, Butidae and Eleotridae, but with very low support (0.52) (S25 Fig in S1 File; note also decreased support for some backbone nodes in this tree).

In the Maximum Parsimony analyses, inclusion of both species could not resolve their phylogenetic position, as in the Bayesian tree, but here the resolution of the backbone of the tree is even more severely reduced (S26 Fig in S1 File). The Maximum Parsimony results for †*Pa. bleicheri* (S27 Fig in S1 File) match those of the Bayesian analyses, whereas when only †*Pa. chapelcorneri* is included, it is placed within Thalasseleotrididae ($< 50\%$, S28 Fig in S1 File).

†*Pirskenius spp. and* †*Paralates spp.*–When all four †*Paralates* and †*Pirskenius* species were included in the Bayesian analysis, they were recovered in a †*Paralates* + †*Pirskenius* clade (0.74), which was resolved as sister to Thalasseleotrididae with moderate support (0.79) (S29 Fig in S1 File). In the Maximum Parsimony tree †*Pirskenius* is recovered monophyletic (64%) but its position and the positions of the two †*Paralates* species within a clade together with Butidae, Thalasseleotrididae and 5brG are not resolved; overall, this tree shows very poor resolution at its backbone (S30 Fig in S1 File).

## Discussion

In this study, we have assembled for the first time a dataset comprising molecular and morphological data for Gobioidei that encompasses both extant and fossil species. This approach was necessary as a phylogenetic analysis of the extant species based solely on their morphological characters could only resolve those clades for which morphological apomorphies are known, i.e. 5brG, Thalasseleotrididae, Thalasseleotrididae+5brG, and Eleotridae [see 15, 28], while Butidae, Odontobutidae and Rhyacichthyidae each were recovered as paraphyletic. The overall objective was to investigate whether a fossil gobioid species can be confidently placed at family level in the tree of the extant Gobioidei using a Bayesian or Maximum Parsimony total-evidence phylogenetic approach. The results reveal mostly well supported placement at family level when a single fossil species is added to the total evidence data set of the extant species, especially in the Bayesian setting.

Five of the fossil species used here had previously been assigned at family level based on a comparative approach: †*Lepidocottus aries* had been placed within Butidae [44], †"*Gobius*" *francofurtanus* and †*Gobius jarosi* had been proposed as members of Gobiidae [46, 100], †*Pirskenius* spp. had been placed in its own family †Pirskeniidae [49] and a sister group relation of †Pirskeniidae to Thalasseleotrididae + 5brG has been suggested [11]. Each of those fossil taxa have been recovered in corresponding positions in the present study (Figs 6B, 7A and 8A). This implies that using comparative morphology has been a very appropriate method to classify those fossils. The family assignment of the remaining five fossil species analyzed here (†*Carlomonnius quasigobius*, †*Eleogobius* spp., †*Paralates* spp.) had been left as *incertae sedis* in previous work because they possess a mosaic set of characters that is not known among extant gobioids [42, 43, 64].

†*Carlomonnius quasigobius* originates from the Eocene of Monte Bolca in northern Italy [64], from the lower to middle Eocene (Ypresian to Lutetian, c. 50–40 Ma, see [101]). It is placed in Butidae in our study (Fig 6A), and it shares with some Butidae (especially with *Kribia*) a very small size, but any comparative approach would not have assigned †*C. quasigobius* to this family because it has only five branchiostegal rays (vs. six in Butidae) and a continuous dorsal fin (vs. divided). The feature that seems to have placed †*C. quasigobius* within the Butidae and close to *Kribia* is the number of 11 branched and segmented caudal fin rays, which is uncommon among other Gobioidei. Nevertheless, given that †*C. quasigobius* is the oldest gobioid species currently known [64], an assignment to the Gobiidae (with which it shares the number of five branchiostegal rays) seems unlikely and its classification within the Butidae appears to be more plausible. It would expand the fossil record of Butidae from the early Oligocene (c. 30 Ma, [102]) to the early-middle Eocene (c. 50–40 Ma). However, an additional possibility is that †*C. quasigobius* is a member of an extinct gobioid family or a "stem gobioid" showing a mixture of derived characters (five branchiostegal rays, dorsal postcleithrum absent, 11 (7+6) segmented and branched caudal-fin rays, four pelvic-fin rays) and plesiomorphic ones (dorsal fin continuous, 24 (10+14) vertebrae, autogenous haemal spine of the second preural centrum, first two abdominal centra shortened, first dorsal-fin pterygiophore inserting in the second interneural space) [64].

In case of †*Eleogobius* and †*Paralates*, the resulting phylogenies indicate that these genera are not monophyletic and their species may not even belong to the same family (see Figs 7B and 8B, S14–S18 Figs and S24–S28 Figs in S1 File). The two species of †*Eleogobius* have been reported from the lower and middle Miocene (c. 17–14 Ma) of Central Europe, specifically southern Germany [42], Austria ([103], as *Gobius*), Switzerland [104, 105], and Croatia [106]. They have been interpreted as belonging to the same genus because they share a T-shaped palatine, absence of an endopterygoid and presence of six branchiostegal rays, and their otoliths are superficially similar, but show differences to recognize the two species [42]. Of those characters, the T-shaped palatine and absence of an endopterygoid can be considered as apomorphic [23, 28], and would support assignment to Gobiidae, which is proposed, based on our phylogenetic results, for †*E. brevis* (Fig 7A). In contrast, the phylogenetic position of †*E. gaudanti* near Thalasseleotrididae (Fig 7B) is difficult to understand as no potential synapomorphies are known that can be recognized in a fossil. However, it is more or less consistent with the hypothesis of Gierl and Reichenbacher [42] that †*Eleogobius* is somewhat "in-between" the 5brG clade and the 6brG gobioids. Furthermore, Bradić-Milinović et al. [41] have recognized a difference in the arrangement of the branchiostegals in the two *Eleogobius* species, which appears to support the possibility that †*Eleogobius* is not monophyletic.

In the case of †*Paralates*, a family assignment had not previously been proposed. †*Paralates bleicheri* has only been found in lower Oligocene deposits of the southern Upper Rhinegraben [40]. An assignment of †*P. bleicheri* to the Odontobutidae, as indicated in our analysis, receives little support based on its skeletal traits as all shared characters with the Odontobutidae represent plesiomorphic character states (e.g. seven spines in the first dorsal fin, 8–9 rays in the second dorsal fin, presence of postmaxillary process) (see S1 Table). Finds of fossil skeletons of †*P. bleicheri* with otoliths preserved in situ would be necessary to reinforce this hypothesis.

†*Paralates chapelcorneri* originates from the upper Eocene "Chapelcorner Fish bed" of southern England (Isle of Wight) [38, 43]. No otoliths have yet been reported from the "Chapelcorner Fish bed". The family assignment of this species remains a topic of future research based on new material of this species. Moreover, a possible relationship between †*Paralates* spp. and †*Pirskenius* spp. was indicated in the Bayesian and Maximum Parsimony trees (S25, S26 Figs in S1 File), which is possibly due to their specific combination of plesiomorphic (e.g.

seven spines in the first dorsal fin, nine anal-fin rays) and apomorphic traits (e.g. 12 abdominal vertebrae, presence of interneural gap).

Our study also yields some new insights from a methodological point of view. Adding morphological data from only extant species to the molecular dataset had practically no influence on the tree topology and support values (S3 Fig in S1 File). Likewise, the inclusion of a single fossil or of two congeneric species did not change the tree topology, only sometimes some support values decreased (Figs 6–8). However, when all fossils were included in the total evidence phylogenetic framework, the resolution of relationships between families and most fossil taxa dramatically collapsed (Fig 5B). Notably, the morphological phylogeny including the extant and all fossil species was less severely collapsed in the backbone of the tree as the 5brG clade and the Thalasseleotrididae were resolved (Fig 5A). It seems that in the case of the total evidence phylogeny the fossil taxa added a high level of conflicting phylogenetic signals, which could not be overcome by the molecular data despite the latter having orders of magnitude more characters and harbouring strong signal for resolving gobioid phylogeny. A possible explanation is that the fossil taxa do not only add morphological information, but also a lot of question marks to the matrix, because, depending on their preservation, some morphological traits cannot be determined. An additional (or alternative) explanation is that many extinct gobioid clades and families, each with a unique character combination, existed in the past [11, 41, 45, 49]. These cannot be 'forced' in the tree of extant species and eventually may also be responsible for the collapse of the molecular backbone of the extant families. This highlights that increased sampling of fossil taxa in a total-evidence context is not universally beneficial, as might be expected, but strongly depends on the study group and peculiarities of the morphological data.

## Conclusion

We have presented a total-evidence dataset comprising molecular and morphological data of 29 extant gobioid species representing all families. Bayesian and Maximum Parsimony analyses revealed that this dataset is sufficient to achieve a molecular 'backbone' that fully conforms to previous molecular work. The new dataset can be used to analyze the family assignment of fossil skeletal-based gobioid species using Bayesian and Maximum Parsimony total-evidence phylogenetic approaches, which has not been possible before.

Our phylogenetic analyses confirmed the family assignment of those fossil gobioid species for which such a placement had been proposed in previous works. It is thus evident that comparative morphology remains an appropriate method to classify some gobioid fossils. However, our phylogenetic analyses could also suggest relationships of fossil gobioid species for cases where the comparative approach did not yield conclusive results. An example is †*Carlomonnius quasigobius*, which is the oldest gobioid fossil to date and our phylogeny suggests that it could be a possible member of the Butidae, which would expand the known age of fossil butids from the early Oligocene (30 Ma) to the early-middle Eocene (40–50 Ma). Although such positioning of †*C. quasigobius* remains somewhat speculative for now, it can give hints to look at certain fossil species from a different and new perspective.

We think that the total evidence framework presented here will be beneficial for all future work dealing with the phylogenetic placement of fossil gobioids and thus will help to improve our understanding of the evolutionary history of these fascinating fishes.

## Supporting information

**S1 Appendix. Short description of the 10 fossil specimens included in this study.**
(DOCX)

**S1 Table. Part A.** Distribution of character states of the characters 1–54 among the examined 29 extant and 10 fossil gobioid species. **Part B.** List of characters, their states and literatur sources. Colour indicates how the character state was determined;? indicates that character state is not known.
(XLSX)

**S1 File. Caption for S1–S30 Figs.** Maximum Parsimony trees with bootstrap values and 50% majority-rule consensus (MRC) Bayesian trees with posterior probabilities based on the different data sets used in this study.
(DOCX)

## Acknowledgments

For providing technical assistance and access to specimens from the SNSB-ZSM collection we thank D. Neumann and U. Schliewen, respectively, (both SNSB-ZSM, Munich, Germany), with special thanks to the latter for help in the acquisition of specimens and insightful discussions. We are grateful to E. Bernard (NHMUK, London, UK), S. Merker (SMNS, Stuttgart, Germany), M. Parrent (MRAC, Tervuren, Belgium), and T. Přikryl (Charles University, Prague, Czech Republic), who all helped to study the specimens kept in the collections of their institutions. Sincere thanks go to W. Schwarzhans (Hamburg, Germany) for providing photographs of the otoliths of *Lesueurigobius sanzi*, *Asterropteryx semipunctata*, *Eucyclogobius newberry*, *Thalasseleotris iota* and *Dormitator maculatus* (shown in Fig 3) and D. Nolf and K. Hoedemakers (both IRSNB, Brussels, Belgium) for supplying the otolith SEM image of *Odontobutis obscurus* (shown in Fig 3). We thank G. Wörheide (LMU Munich, Germany) for providing access to computational resources. Finally we thank the reviewers and the Academic Editor of PLOS One for their constructive and valuable remarks, which greatly helped to improve the manuscript.

## Author Contributions

**Conceptualization:** Christoph Gierl, Bettina Reichenbacher.

**Data curation:** Christoph Gierl, Bettina Reichenbacher.

**Formal analysis:** Christoph Gierl, Martin Dohrmann, Bettina Reichenbacher.

**Investigation:** Christoph Gierl, Bettina Reichenbacher.

**Methodology:** Christoph Gierl, Martin Dohrmann, Bettina Reichenbacher.

**Resources:** Philippe Keith, William Humphreys, Hamid R. Esmaeili, Jasna Vukić, Radek Šanda, Bettina Reichenbacher.

**Validation:** Christoph Gierl, Martin Dohrmann, Bettina Reichenbacher.

**Visualization:** Christoph Gierl, Martin Dohrmann, Bettina Reichenbacher.

**Writing – original draft:** Christoph Gierl, Martin Dohrmann, Bettina Reichenbacher.

**Writing – review & editing:** Christoph Gierl, Martin Dohrmann, Philippe Keith, William Humphreys, Hamid R. Esmaeili, Jasna Vukić, Radek Šanda, Bettina Reichenbacher.

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
