## [Decision Letter · Decision Letter 0]

7 Jan 2022

PONE-D-21-29896An integrative phylogenetic approach for inferring relationships of fossil gobioids (Teleostei: Gobiiformes)PLOS ONE

Dear Dr. Reichenbacher,

Thank you for submitting your manuscript to PLOS ONE. After careful consideration, we feel that it has merit but does not fully meet PLOS ONE’s publication criteria as it currently stands. Therefore, we invite you to submit a revised version of the manuscript that addresses the points raised during the review process.

Note that reviewer #2 didn't agree with some familial assignations of fossil species but also he/she didn't take into account one of the supplementary files I received (and sent to her/him) after the submission of the manuscript. The support of the familial position of these species are the main reason by which I followed the "major revision" suggestion by that reviewer.The reviewer #2 also recommends to follow Paul Sereno's definitions for morphological characters, which I'll leave to the criterion of the authors (I wouldn't consider it necessarily an improvement).

We look forward to receiving your revised manuscript.

Kind regards,

Juan Marcos Mirande

Academic Editor

PLOS ONE

https://journals.plos.org/plosone/s/file?id=ba62/PLOSOne_formatting_sample_title_authors_affiliations.pdf”

2. To comply with PLOS ONE submissions requirements, please provide methods of sacrifice in the Methods section of your manuscript.

3. In your manuscript, please provide additional information regarding the specimens used in your study. Ensure that you have reported specimen numbers and complete repository information, including museum name and geographic location.

For more information on PLOS ONE's requirements for paleontology and archaeology research, see https://journals.plos.org/plosone/s/submission-guidelines#loc-paleontology-and-archaeology-research.

“For providing technical assistance and access to specimens from the SNSB-ZSM collection we thank D. Neumann and U. Schliewen, respectively, (both SNSB-ZSM, Munich, Germany), with special thanks to the latter for help in the acquisition of specimens and insightful discussions. We are grateful to E. Bernard (NHMUK, London, UK), S. Merker (SMNS, Stuttgart, Germany), M. Parrent (MRAC, Tervuren, Belgium), and T. Přikryl (Charles University, Prague, Czech Republic), who all helped to study the specimens kept in the collections of their institutions. Sincere thanks go to W. Schwarzhans (Hamburg, Germany) for providing photographs of the otoliths of Lesueurigobius sanzi, Asterropteryx semipunctata and Dormitator maculatus (shown in Fig 3) and D. Nolf and K. Hoedemakers (both IRSNB, Brussels, Belgium) for supplying the otolith SEM image of Odontobutis obscurus (shown in Fig 3). For Rhyacichthys we thank the Vanuatu Environment Unit (Permit Numbers ENV326/001/1/07/DK and ENV326/001/1/08/D) and for Protogobius the New Caledonian South Province for allowing sampling (Permit Number 1224-08/PS). We thank G. Wörheide (LMU Munich, Germany) for providing access to computational resources. We acknowledge funding for this project from the Deutsche Forschungsgemeinschaft to BR (RE-1113/20), and from the Ministry of Culture of the Czech Republic to RS (DKRVO 2019-2023/6.III.d National Museum, 00023272).”

“BR received funding from the Deutsche Forschungsgemeinschaft (grant number RE-1113/20). RS received funding from the Ministry of Culture of the Czech Republic (grant number DKRVO 2019-2023/6.III.d National Museum, 00023272).

https://www.dfg.de/

https://www.mkcr.cz/

Reviewers' comments:

Reviewer's Responses to Questions

**Comments to the Author**

1. Is the manuscript technically sound, and do the data support the conclusions?

Reviewer #1: Partly

Reviewer #2: Partly

2. Has the statistical analysis been performed appropriately and rigorously? 

Reviewer #1: Yes

Reviewer #2: Yes

3. Have the authors made all data underlying the findings in their manuscript fully available?

Reviewer #1: Yes

Reviewer #2: No

4. Is the manuscript presented in an intelligible fashion and written in standard English?

Reviewer #1: Yes

Reviewer #2: Yes

5. Review Comments to the Author

Reviewer #1: Dear authors, This is an interesting article with a novel approach and incorporating otolith research for the first time in a total evidence phylogenetic analysis. My congratulations to your work.

There are, however, a few issues that need to be addressed in my view before this article can be published. I have made some comments in the main text and the supplementary data as attached, but there are a few principle issues which I would like to bring to your attention as follows:

1. I am lacking a discussion about the state of the characters used, i.e., plesiomorphic or apomorphic and polarity. This is particularly so for the otolith characters while for the other characters possibly references could be cited. In general, however, this is my most concerning issue.

2. Did the phylogenetic analyses reflect plesiomorphic versus apomorphic character states and if so, how?

3. The phylogeny collapsed when all fossil taxa where included. I find this intriguing and the discussion to this aspect way too short, while the technical phylogenetic discussion from the various runs is rather exhaustive. Definitely, this aspect requires more attention and consideration and discussion than you provided in the manuscript.

4. Much reference is made to the earliest gobioid on the fossil record- Carlomonnius quasigobius. A very detailed anatomical and phylogenetic analysis was provided by Bannikov & Carnevale (2016) who described it resulting in not placing it in a specific family because of its unique mixture of plesiomorphic and apomorphine characters. In this article it is now based in Butidae based on your phylogenetic modeling alone without discussion of how the results of Bannikov & Carnevale would relate. This needs to be considered and discussed in more detail because of its phylogenetic impact.

With these aspects being carefully addressed I am of the opinion that the manuscript will become a very valuable article both in terms of the methodology applied as well as the results presented.

Best regards,

Werner Schwarzhans

Reviewer #2: This paper is well-written and I appreciate the hard work done by the authors. However, several questions and concerns arose during the readings. The authors claimed that they were able to assign several of the incertae sedis fossils to family level using their total-evidence phylogeny, but in many cases the results are not solid. Also, I was not able to review the Suppl. Figs 1-8 (what I was able to download was a docx. with the figures in the main text, not Suppl. Figs 1-8), so basically, I was only able to access half of the results presented.

1. The step-wise approach (L. 229) seems to have good resolution on the results, but what’s the rationale here to this approach only? Because when multiple fossils are included in the analysis, the phylogeny collapsed (as in L. 664), which can be easily challenged by future works when more evidence and specimens are available, then, the total-evidence phylogeny produced here is not a stable one, and means that you need other robust phylogeny (topology) for the ten fossil taxa (and perhaps not yours). Or, on the contrary, the morphological data are too few to reconstruct a good tree for fossils, leaving those inc. sed. still incertae sedis. If that’s so, then you are forcing the taxonomic assignment to any of the extant families while they might well belong to any of an extinct clade.

2. L. 618, you mentioned these morphological traits, but these are conflicting your assignment below! Therefore, the statement in L. 615-624 is weak, there should be morphological evidence supporting your assignment rather than your phylogenetic tree (L. 583, with low support value!) or just stating the stratigraphic occurrences. Also, if C. quasigobius really is a Butidae, then the expansion of the fossil record is also a big deal (some 10-20 Ma earlier).

3. In my opinion, we only have a limited number of extant goby families and that might not be ideal to place the unknown, yet to be fully discovered fossil taxa to the living scheme, as there are myriads of unknown diversity behind (as you wrote in L. 611-614). Therefore, it might not be so surprising that the results are not well resolved as all.

4. L. 218, you mentioned some molecular data came from “Thacker et al. [62]”, but it didn’t match the ref. 62 in the References.

5. Suppl. Table. I suggest following Sereno (2007) for character description.

Sereno, C. P. 2007. Logical basis for morphological characters in phylogenetics. Cladistics 23(6): 565-587.

6. PLOS authors have the option to publish the peer review history of their article (what does this mean?). If published, this will include your full peer review and any attached files.

Reviewer #1: **Yes: **Werner Schwarzhans

Reviewer #2: No

---

## [Author Response · Author response to Decision Letter 0]

11 Apr 2022

Responses to specific reviewer comments are provided in the file "Response to Reviewers".

Response to the comments of the Editor are also included in the file "Response to Reviewers".

---

## [Decision Letter · Decision Letter 1]

17 Jun 2022

PONE-D-21-29896R1An integrative phylogenetic approach for inferring relationships of fossil gobioids (Teleostei: Gobiiformes)PLOS ONE

Dear Dr. Reichenbacher,

Thank you for submitting your manuscript to PLOS ONE. The manuscript is almost ready to be sent to production.  I just think the comments of reviewer #1 about the abstract could be addressed by the authors before publishing. I selected Minor Revision just to give the authors the chance to modify it previous to the Galley Proofs.

We look forward to receiving your revised manuscript.

Kind regards,

Juan Marcos Mirande

Academic Editor

PLOS ONE

Journal Requirements:

Reviewers' comments:

Reviewer's Responses to Questions

**Comments to the Author**

1. If the authors have adequately addressed your comments raised in a previous round of review and you feel that this manuscript is now acceptable for publication, you may indicate that here to bypass the “Comments to the Author” section, enter your conflict of interest statement in the “Confidential to Editor” section, and submit your "Accept" recommendation.

Reviewer #1: All comments have been addressed

Reviewer #2: All comments have been addressed

2. Is the manuscript technically sound, and do the data support the conclusions?

Reviewer #1: Yes

Reviewer #2: (No Response)

3. Has the statistical analysis been performed appropriately and rigorously? 

Reviewer #1: Yes

Reviewer #2: (No Response)

4. Have the authors made all data underlying the findings in their manuscript fully available?

Reviewer #1: Yes

Reviewer #2: (No Response)

5. Is the manuscript presented in an intelligible fashion and written in standard English?

Reviewer #1: Yes

Reviewer #2: (No Response)

6. Review Comments to the Author

Reviewer #1: Thank you for considering all the comments and suggestions made. In my opinion, the manuscript reads very well and conclusive now. Two minor aspects I would like to bring to your attention before final submission and publication:

1.- In the abstract some sort of reference should be made to the problems concerning the tree collapsing when including all fossil taxa.

2.- In the phylogenetic analysis you mention the problem of the tree collapsing with all fossil data for the first time. Later, in the chapter Discussion you have outlined your thoughts to this effect, which in my opinion are quite comprehensive. You should refer to this discussion where you mention the effect first in the text as a guidance for the reader.

Looking forward to see you study published and best wishes,

Werner Schwarzhans

Reviewer #2: Dear authors,

After reviewing the revised manuscript, I judge that all my previous comments have been addressed adequately. My congratulations to your work.

Best,

Chien-Hsiang

7. PLOS authors have the option to publish the peer review history of their article (what does this mean?). If published, this will include your full peer review and any attached files.

Reviewer #1: **Yes: **Werner Schwarzhans

Reviewer #2: **Yes: **Chien-Hsiang Lin

---

## [Author Response · Author response to Decision Letter 1]

20 Jun 2022

We have included all suggested comments in the revised manuscript.

---

## [Editor Report · Decision Letter 2]

24 Jun 2022

An integrative phylogenetic approach for inferring relationships of fossil gobioids (Teleostei: Gobiiformes)

PONE-D-21-29896R2

Dear Dr. Reichenbacher,

We’re pleased to inform you that your manuscript has been judged scientifically suitable for publication and will be formally accepted for publication once it meets all outstanding technical requirements.

Kind regards,

Juan Marcos Mirande

Academic Editor

PLOS ONE
---

## [Editor Report · Acceptance letter]

30 Jun 2022

PONE-D-21-29896R2 

An integrative phylogenetic approach for inferring relationships of fossil gobioids (Teleostei: Gobiiformes) 

Dear Dr. Reichenbacher:

I'm pleased to inform you that your manuscript has been deemed suitable for publication in PLOS ONE. Congratulations! Your manuscript is now with our production department. 

Kind regards, 

on behalf of

Dr. Juan Marcos Mirande 

Academic Editor

PLOS ONE